# PanoGRF: Generalizable Spherical Radiance Fields for Wide-baseline Panoramas

Zheng Chen[1]*, Yan-Pei Cao[2], Yuan-Chen Guo[1], Chen Wang[1], Ying Shan[2], Song-Hai Zhang[1]
[1]Tsinghua University [2]ARC Lab, Tencent PCG
China
{chenz20,guoyc19}@mails.tsinghua.edu.cn caoyanpei@gmail.com
cw.chenwang@outlook.com yingsshan@tencent.com shz@tsinghua.edu.cn

## Abstract

Achieving an immersive experience enabling users to explore virtual environments with six degrees of freedom (6DoF) is essential for various applications such as virtual reality (VR). Wide-baseline panoramas are commonly used in these applications to reduce network bandwidth and storage requirements. However, synthesizing novel views from these panoramas remains a key challenge. Although existing neural radiance field methods can produce photorealistic views under narrow-baseline and dense image captures, they tend to overfit the training views when dealing with *wide-baseline* panoramas due to the difficulty in learning accurate geometry from sparse 360° views. To address this problem, we propose PanoGRF, Generalizable Spherical Radiance Fields for Wide-baseline Panoramas, which construct spherical radiance fields incorporating 360° scene priors. Unlike generalizable radiance fields trained on perspective images, PanoGRF avoids the information loss from panorama-to-perspective conversion and directly aggregates geometry and appearance features of 3D sample points from each panoramic view based on spherical projection. Moreover, as some regions of the panorama are only visible from one view while invisible from others under wide baseline settings, PanoGRF incorporates 360° monocular depth priors into spherical depth estimation to improve the geometry features. Experimental results on multiple panoramic datasets demonstrate that PanoGRF significantly outperforms state-of-the-art generalizable view synthesis methods for wide-baseline panoramas (e.g., OmniSyn) and perspective images (e.g., IBRNet, NeuRay). Poject Page: https://thucz.github.io/PanoGRF/.

## 1 Introduction

The rise of 360° cameras and virtual reality headsets has fueled the popularity of 360° images among photographers and tourists. Commercial VR platforms, such as Matterport [2], enable users to experience virtual walks in 360° scenes by interpolating between panoramas [17]. Wide-baseline panoramas are frequently employed on these platforms for capture and network transmission to reduce storage space and bandwidth requirements. Consequently, synthesizing novel views from wide-baseline panoramas is an essential task for providing a seamless six degrees of freedom (6DoF) experience to users.

However, synthesizing novel views from a pair of wide baseline panoramas encounters two primary challenges. *First*, the input views are sparse, causing existing state-of-the-art methods such as NeRF [21] to struggle in learning accurate geometry due to the shape-radiance ambiguity [37],

---

*Work done during an internship at ARC Lab, Tencent PCG
[2]Matterport: https://matterport.com

37th Conference on Neural Information Processing Systems (NeurIPS 2023).

leading to overfitting of the input views. *Second*, certain regions in the scene may only be visible in one view but not in another, making it challenging to provide a correct geometry prior to NeRF using only multi-view stereo. State-of-the-art generalizable NeRF methods [35, 32, 20] address the overfitting problem by incorporating scene priors into NeRF; however, they are designed for perspective images and require conversion of panoramas into perspective views when applied, resulting in information loss and suboptimal performance due to the limited field of view. Although spherical radiance fields [11] have been proposed to render panoramic images based on spherical projection, they still suffer from the overfitting problem without considering the 360° scene prior. Other approaches designed for 360° view synthesis, such as multi-sphere images (MSI) [2] and mesh-based methods [17], exhibit limited expressiveness and rendering quality compared to NeRF due to the use of discretized scene representations.

In this work, we present a method that addresses the overfitting issues in spherical radiance fields by incorporating 360° scene priors. Unlike existing generalizable methods that require panorama-to-perspective conversion, our approach retains the panoramic representation. Furthermore, we incorporate 360° monocular depth to alleviate the view occlusion problem.

To address the first challenge, we present a solution in the form of *generalizable spherical radiance fields*. To render the panorama at a new position, spherical radiance fields (Spherical NeRF) cast a ray for each pixel using spherical projection, sample points along the ray, and aggregate the colors of these points based on their density values following NeRF [21]. In our method, named PanoGRF, we incorporate 360° scene priors into Spherical NeRF. Specifically, we extract appearance and geometry features from input panoramas and estimated spherical depths through convolutions, respectively. PanoGRF accurately aligns the queried 3D points with the corresponding pixels in panoramas using spherical projection. This alignment strategy leverages the full field-of-view characteristic of panoramas and eliminates the information loss of panorama-to-perspective conversion. The local appearance and geometry features at these corresponding pixels are then aggregated and serve as the conditional input for Spherical NeRF.

To tackle the second challenge, we enhance the accuracy of depth estimation in 360° multi-view stereo by integrating a 360° monocular depth estimation network. In 360° multi-view stereo, depth inaccuracies can arise due to view inconsistencies caused by occluded regions. Inspired by the perspective method proposed in [3], we sample depth candidates around the estimated 360° monocular depth, assuming a Gaussian distribution. These depth candidates are then utilized in sphere sweeps during 360° multi-view matching. By incorporating monocular depth candidates, we can improve the accuracy of depth estimation in regions with view inconsistencies. This improvement contributes to the robustness of the geometry features employed in PanoGRF and ultimately leads to superior view synthesis performance.

In summary, we make the following contributions: 1) We propose the design of generalizable spherical radiance fields to learn 360° scene priors that alleviate the overfitting problem when dealing with wide-baseline panoramas. 2) We incorporate 360° monocular depth into the spherical multi-view stereo to mitigate the view inconsistency problem caused by occluded regions, which results in more robust geometry features, ultimately improving the rendering performance. 3) We achieve the state-of-the-art novel view synthesis performance on Matterport3D [4], Replica [28] and Residential [8] under the wide-baseline setting.

## 2 Related Work

In this section, we briefly review the methods of perspective view synthesis and 360° view synthesis.

### 2.1 Perspective View Synthesis

Recently, NeRF (neural radiance fields [21]) has become the mainstream scene representation due to its photo-realistic rendering quality for perspective view synthesis. However, given sparse input views, NeRF's per-scene optimization approach is prone to overfitting the training views and fails to learn the correct geometry due to the shape-radiance ambiguity, leading to significant floaters in novel views. Several subsequent methods [12, 6, 15, 22, 5, 34] have attempted to address this issue by incorporating depth constraints from COLMAP [26] or other regularization terms, but they still rely on per-scene optimization and lack the ability to generalize to new scenes. In contrast,

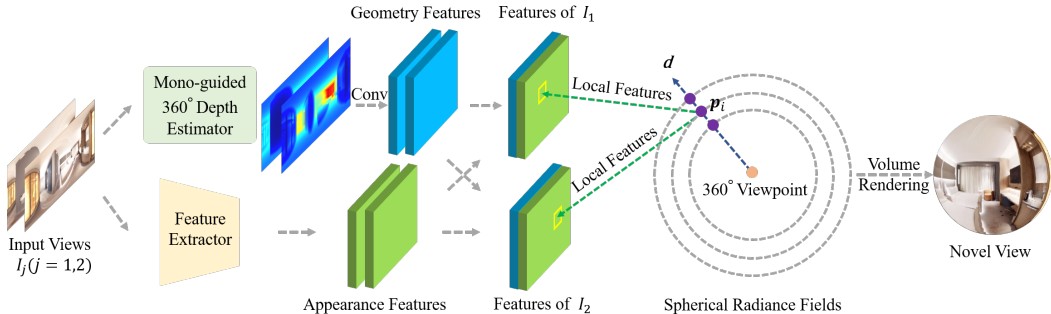

Figure 1: The overview of PanoGRF. Initially, we employ convolutional neural networks (CNNs) to extract appearance features and geometry features from the input views. The geometry feature is obtained from the predicted spherical depth generated by our proposed Mono-guided $360°$ Depth Estimator (Sec. 3.3). Next, we cast rays based on spherical projection to render a novel $360°$ view (Sec. 3.1). Along each ray, every 3D sample point $p_i$ is projected onto the corresponding panoramic grid coordinate $(u, v)$. The local geometry feature $g_{i,j}$ at $(u, v)$ is decoded into $v_{i,j}$ (the visible probability of $p_i$ for the $j$-th view). The local appearance feature $f_{i,j}$ at $(u, v)$ is aggregated into $f_i$ along with $v_{i,j}$ (Sec. 3.2). Subsequently, the aggregated feature $f_i$ is decoded to determine the color and density of $p_i$. Finally, a novel $360°$ view is synthesized through volume rendering (Sec. 3.1).

PixelNeRF [35], GRF [30] and IBRNet [32] condition NeRF rendering with pixel-aligned appearance features to learn scene priors from large-scale datasets, enabling direct inference with input images. NeuRay [20] aggregates visibility information (obtained from multi-view stereo) and appearance features from each view to generate the density and color of 3D sample points. However, when dealing with panoramas, these methods require converting the panoramas to perspective images, leading to information loss and suboptimal performance due to limited field-of-view in each perspective view. In contrast, PanoGRF operates directly on panoramas using spherical projection, eliminating the need for the panorama-to-perspective conversion.

## 2.2 $360°$ View Synthesis

Multi-sphere images (MSI) are widely adopted for $360°$ view synthesis. It is introduced by [2] to represent scenes with the input of omnidirectional stereo video. SOMSI [8] improved the expressive ability of MSI by incorporating high-dimensional feature layers and achieved real-time performance. However, MSI's capability is limited to narrow-baseline scenarios due to the back surface rendering issues as discussed in [18], which attempted to expand the renderable viewpoint range of MSI through the interpolation of two MSI instances. But their pure convolutional architecture only works at low resolutions. OmniSyn [17] constructs two meshes for input panoramas using $360°$ multi-view stereo. It warps the meshes to the target view and fuses the colors attached to the meshes with an in-painting network. This approach frequently produces ghosting artifacts due to surface-based warping and blending. $360°$ Roam [11] divides NeRF into multiple blocks to achieve real-time rendering of panoramas, akin to KiloNeRF [23]. But it fits a single scene without considering scene priors, making it unsuitable for wide-baseline panoramas. In contrast to the aforementioned methods, we introduce generalizable spherical radiance fields to learn both geometric and appearance priors from $360°$ datasets, enabling better generalization to unseen $360°$ scenes.

However, $360°$ multi-view stereo alone cannot provide a robust geometric prior, as it fails to address the occlusion issue in the wide-baseline setting. Recently, numerous researchers have focused on $360°$ monocular depth estimation [13, 19, 1] to mitigate spherical distortion by harnessing the fusion of equirectangular and perspective projections. Inspired by MaGNet [3], we employ $360°$ monocular depth prior to guide the construction of $360°$ cost volume, which enhances the quality of geometric features and boosts rendering performance.

# 3 Method

PanoGRF facilitates novel view synthesis from wide-baseline panoramas as shown in Fig. 1. Before introducing PanoGRF, we first review NeRF and its spherical variant in Sec. 3.1. We demonstrate the two important parts of PanoGRF, the generalizable spherical radiance fields, and the mono-guided spherical depth estimator in Sec. 3.2 and Sec. 3.3 respectively.

## 3.1 Spherical Radiance Fields (Spherical NeRF)

To render a pixel, NeRF [21] casts a ray from a given viewpoint and parameterizes the ray as $\boldsymbol{p}(t) = \boldsymbol{o} + t\boldsymbol{d}, t \in \mathbb{R}^+$, where $\boldsymbol{o}$ is the camera center and $\boldsymbol{d}$ is the ray direction. $N$ points are sampled along the ray with increasing $t$, we denote $\boldsymbol{p}_i \equiv \boldsymbol{p}(t_i), i = 1, ..., N$. NeRF utilizes an MLP to map the position $\boldsymbol{p}_i$ together with viewing direction $\boldsymbol{d}$ into color $\boldsymbol{c}_i$ and density $\sigma_i$. The pixel color of the ray $\boldsymbol{p}$ can then be computed by:

$$\boldsymbol{c} = \sum_{i=1}^{N} w_i \boldsymbol{c}_i, \tag{1}$$

where $\boldsymbol{c} \in \mathbb{R}^3$ is the rendered color of $\boldsymbol{p}$, and $\boldsymbol{c}_i \in \mathbb{R}^3$ is the color of the sampled point $\boldsymbol{p}_i$. $w_i$ is the blending weight of the point $\boldsymbol{p}_i$, which indicates the probability that a ray travels from the origin to $t_i$ without hitting any particle and terminates in the range $(t_i, t_{i+1}]$. It is computed by $w_i = \prod_{k=1}^{i-1}(1 - \alpha_k)\alpha_i$, where $\alpha_i$ is the alpha values in the depth range $(t_i, t_{i+1}]$, $\alpha_i = 1 - \exp(-\sigma_i \delta_i)$ and $\delta_i = t_{i+1} - t_i$. The color $\boldsymbol{c}$ of a ray $\boldsymbol{p}$ is optimized by a photometric loss:

$$\mathcal{L} = \sum \|\boldsymbol{c} - \boldsymbol{c}_{gt}\|^2, \tag{2}$$

where $\boldsymbol{c}_{gt}$ represents the ground truth color.

**Ray-casting based on spherical projection**    The panorama uses the panoramic pixel grid coordinate system, while the coordinate systems in NeRF are all Cartesian. By employing *spherical projection* [8, 11], a pixel $(u, v)$ in the panorama is firstly transformed into spherical polar coordinate $(\phi, \theta)$ and subsequently into cartesian coordinate $(x, y, z)$. Detailed transformation formulas can be seen in the supplementary material. The ray direction emitted from the pixel $(u, v)$ can be computed by $\boldsymbol{d} = \mathbf{R}[x, y, z]^T$, where $\mathbf{R} \in \mathbb{R}^{3 \times 3}$ is the panorama's camera-to-world rotation matrix.

## 3.2 Generalizable Spherical Radiance Fields

Under a wide baseline, Spherical NeRF tends to overfit training views and struggle to generate plausible novel views. To address this limitation, we draw inspiration from the generalizable NeRF approaches [35, 32, 20] designed for perspective images. In our work, we propose to incorporate $360°$ scene priors into Spherical NeRF by aggregating local features from input panoramas.

**Alignment based on spherical projection**    Panoramas can be converted into perspective images, and then generalizable NeRF methods for perspective images can be applied to novel $360°$ view synthesis. However, when projecting the 3D sample point $\boldsymbol{p}_i$ onto the source perspective view, it may fall outside the image borders or even be located behind the source perspective camera (with a $z$-depth < 0) due to the limited field-of-view. This can introduce errors in the aggregation of features and result in poor rendering results. To overcome this problem, we directly align $\boldsymbol{p}_i$ with the corresponding pixels in panoramas using spherical projection, enabling us to leverage the full field-of-view characteristic of panoramas. We first transform $\boldsymbol{p}_i$ into the camera's Cartesian coordinate system $(x, y, z)$ of the $j$-th panoramic view $I_j$. Subsequently, $(x, y, z)$ is converted into spherical polar coordinate $(\theta, \phi, t)$ ($t \in \mathbb{R}^+$ indicates the spherical depth of $\boldsymbol{p}_i$ for $I_j$) and finally transformed into the panoramic grid coordinate $(u, v)$. The specific formulas for these transformations can be found in the supplementary material.

**Appearance and geometry feature aggregation**    We follow NeuRay [20] to aggregate appearance and geometry features. But differently, we take panoramas and spherical depth (radial distance from the camera center) as input instead of perspective images and $z$-depth. PanoGRF respectively extracts appearance feature $\mathbf{W}_j$ from the $j$-th panoramic view ($j = 1, 2$) and geometry feature $\mathbf{G}_j$ from

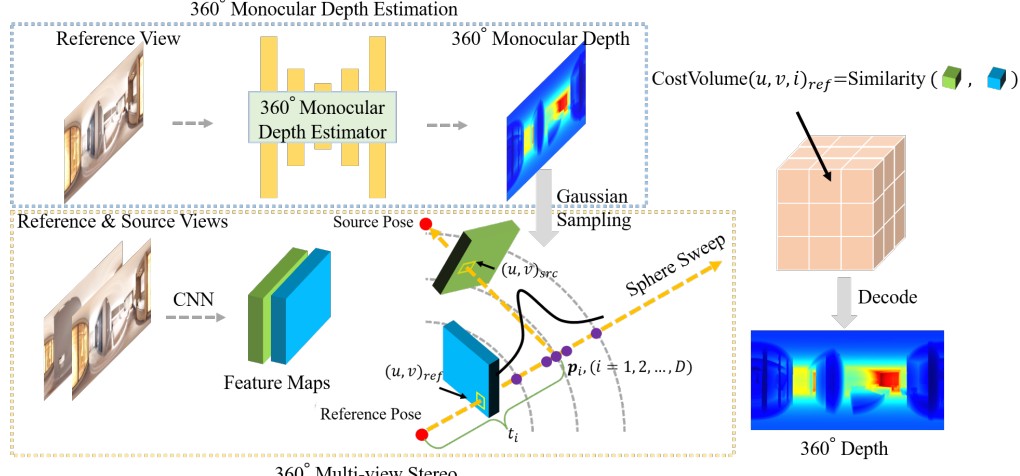

Figure 2: The process of mono-guided 360° depth estimation. We first extract the image features of reference and source views with convolutions. Using spherical projection, we determine the corresponding pixel $(u,v)_{src}$ in the source view for each pixel $(u,v)_{ref}$ of the reference view, with the depth hypothesis of $t_i$. The similarity between the local features at $(u,v)_{ref}$ and $(u,v)_{src}$ is computed as the value of the cost volume at $(u,v,i)$. Depth sampling of $t_i$ is guided by the 360° monocular depth using a Gaussian distribution assumption. The cost volume is obtained after $D$ sphere sweeps. Lastly, the cost volume is decoded into 360° depth using convolutions.

the spherical depth using convolutions. Details for getting the spherical depth will be introduced in Sec. 3.3. For a given 3D sample point $\boldsymbol{p}_i$, we project it onto the panoramic grid coordinate $(u,v)$, and extract the local appearance feature $\boldsymbol{f}_{i,j} = \mathbf{W}_j(u,v)$ and the local geometry feature $\boldsymbol{g}_{i,j} = \mathbf{G}_j(u,v)$ at $(u,v)$. The local geometry feature $\boldsymbol{g}_{i,j}$ is decoded into the visibility $v_{i,j}$ (demonstrated below) using an MLP. The local appearance feature $\boldsymbol{f}_{i,j}$ and visibility $v_{i,j}$ of the $j$-th panoramic view are aggregated for $\boldsymbol{p}_i$ with an aggregation network $\mathcal{F}$:

$$\boldsymbol{f}_i = \mathcal{F}(\{\boldsymbol{f}_{i,j}, v_{i,j} | j = 1, 2\}). \tag{3}$$

According to the local geometry feature $\boldsymbol{g}$ of an input panorama, we predict the visibility function $v(t)$ to indicate the probability that a point at spherical depth $t$ (instead of $z$-depth in NeuRay) is visible for the input panorama. $v(t)$ ($v(t) \in [0,1]$) is represented as $v(t) = 1 - o(t)$, where $o(t)$ is the occlusion probability. To parameterize $o(t)$, we employ a mixture of $N_l$ logistic distributions:

$$o(t; \mu_k, \sigma_k, m_k) = \sum_{k}^{N_l} m_k S((t - \mu_k)/\sigma_k), \tag{4}$$

where $\mu_k$, $\sigma_k$ and $m_k$ are the mean, standard variance, and blending weight of the $k$-th logistic distribution respectively. $\sum_{i}^{N_l} m_k = 1$ and $S(\cdot)$ denotes a sigmoid function. The parameters $[\mu_k, \sigma_k, m_k]$ are decoded from the local geometry feature vector $\boldsymbol{g}$ by an MLP. For each 3D sample point $\boldsymbol{p}_i$, we compute its spherical depth $t_i$ for $j$-th input view based on spherical projection and obtain the visible probability $v_{i,j} = v_j(t_i)$ of $\boldsymbol{p}_i$ for the $j$-th panoramic view. Then we aggregate $v_{i,j}$ into $\boldsymbol{f}_i$ by Eq. 3 and decode $\boldsymbol{f}_i$ into the color and density of $\boldsymbol{p}_i$.

### 3.3 Mono-guided Spherical Depth Estimator

To obtain the visibility in Sec. 3.2, we need to predict the spherical depth of each input view. The process is illustrated in Fig. 2.

360° **multi-view stereo (MVS)**   With the input of multi-view panoramas, we estimate the spherical depth of the reference view using 360° multi-view stereo similar to previous methods [17, 14]. Firstly, we extract the image features of the reference and source views with an image encoder. For each pixel $(u,v)_{ref}$ of the reference view, assuming its depth is $t_i$, we find its corresponding pixel $(u,v)_{src}$ in

the source view according to spherical projection. We uniformly sample $D$ depth candidates for $t_i$ (with $i = 1, 2, ..., D$) without considering the monocular depth prior, which will be introduced later. Next, we compute the similarity between the local features at $(u, v)_{ref}$ and $(u, v)_{src}$, considering it as the value of the cost volume at $(u, v, i)$. Assuming the length of the local feature vector is $F$, this process results in a 4D cost volume $\mathbf{V} \in \mathbb{R}^{\frac{H}{4} \times \frac{W}{4} \times D \times F}$ after $D$ sphere sweeps, where $\frac{H}{4}$ and $\frac{W}{4}$ respectively represent the height and width of the feature maps for the reference view. Subsequently, the dimension of the cost volume is reduced to $\frac{H}{4} \times \frac{W}{4} \times D$ through a 3D CNN. Lastly, the processed cost volume is decoded into $360°$ depth using several convolution layers.

**Mono-guided spherical depth sampling** Under the wide-baseline setting, some areas might be visible from one view but occluded from another, in which case $360°$ MVS may struggle to produce accurate depth estimates. To address this challenge, we leverage the $360°$ monocular depth [13] to guide the depth sampling of $360°$ MVS, inspired by perspective methods [36, 31, 3].

For each pixel $(u, v)$ in the panorama, we denote its estimated spherical depth from the monocular depth network as $\mu_{u,v}$. We generate depth candidates in the vicinity of the monocular depth using a Gaussian distribution assumption. Specifically, a search space for each pixel in the panorama is defined as $[\mu_{u,v} - \beta\sigma, \mu_{u,v} + \beta\sigma]$ where $\beta$ and $\sigma$ serve as hyperparameters and $\sigma$ represents the standard deviation. This search space is divided into $N_{mono}$ bins, ensuring that each bin has the same probability mass. We select the midpoint of each bin as the depth candidate. Consequently, the $k$-th depth candidate for pixel $(u, v)$ is defined as $t_{u,v,k} = \mu_{u,v} + b_k\sigma, k = 1, 2, ..., N_{mono}$. $b_k$ represents the offset corresponding to the $k$-th bin. Similar to [3], we calculate $b_k$ as the following:

$$b_k = \frac{1}{2} \left[ \Phi^{-1}(\frac{k-1}{N_{mono}}P^* + \frac{1-P^*}{2}) + \Phi^{-1}(\frac{k}{N_{mono}}P^* + \frac{1-P^*}{2}) \right], \tag{5}$$

where $P^* = \text{erf}(\frac{\beta}{\sqrt{2}})$ denotes the probability mass covered by the search space $[\mu_{u,v} - \beta\sigma, \mu_{u,v} + \beta\sigma]$, $\Phi^{-1}(\cdot)$ is the quantile function of standard normal distribution, $\text{erf}$ is the error function. Considering errors may occur in the $360°$ monocular depth, we use a mixture of $N_{mono}$ monocular depth candidates and $N_{uni}$ uniform depth candidates sampled from a uniform distribution. By constructing $D = N_{uni} + N_{mono}$ sphere sweeps, we obtain a new spherical cost volume. The monocular depth guidance enables us to extract more reliable geometry features employed in PanoGRF.

## 4 Experiment

### 4.1 Metrics and Datasets

PSNR, SSIM [10], LPIPS [38] and WS-PSNR [29] are used as evaluation metrics. We conduct experiments on Matterport3D [4], Replica [28], and Residential [8]. For Matterport3D and Replica, we leverage HabitatAPI [25] to generate the $256 \times 256$ perspective images (representing the six sides of cube-maps) and stitch them into a $512 \times 1024$ panorama. We use panoramic sequences with a length of 3. The middle view is rendered based on the first and last views. We conduct comparative experiments on Matterport3D under fixed camera baselines of 1.0, 1.5, and 2.0 meters, where the camera baseline refers to the distance between the camera centers of the first and last views. The baselines used for Residential and Replica are approximately 0.3 and 1.0 meters, respectively.

### 4.2 Implementation Details

We set $N_{mono} = 5$, $N_{uni} = 59$, and $N_l = 2$. $D = N_{mono} + N_{uni}$ is 64. $\sigma$ used in mono-guided depth sampling is set to 0.5 and $\beta$ is set to 3. Additional details regarding network architecture can be found in the supplementary material.

### 4.3 Comparisons

We compared PanoGRF with S-NeRF (spherical variant of NeRF [21]), IBRNet [32], NeuRay [20], and OmniSyn [17]. We trained S-NeRF from scratch, as it is a per-scene optimization method. Other methods are pre-trained on Matterport3D and tested on unseen testing scenes. We fed the original cube maps of panoramas (in Matterport3D and Replica) to the perspective methods (IBRNet and NeuRay). As there are only panoramas in ERP (equirectangular projection) format for Residential,

cube maps were converted from panoramas. To render a panoramic view for evaluation, we cast rays in IBRNet and NeuRay with spherical projection (refer to Sec. 3.1) and aggregate features with their original perspective projection. We use NeuRay* and IBRNet* to denote variants of NeuRay and IBRNet which render panoramas. We did not compare with methods based on multi-sphere images (MSI) [2, 8] as they can only render novel views within the smallest sphere of MSI, which are unsuitable for wide-baseline panoramas. In the supplementary materials, we provide additional analysis on per-scene fine-tuning of PanoGRF and also compare it with Cross Attention Renderer [7].

Table 1: Quantitative comparisons with baseline methods on Matterport3D. The best results are in bold.

| baseline | 1.0m | | | 1.5m | | | 2.0m | | |
|---|---|---|---|---|---|---|---|---|---|
| method | WS-PSNR↑ | SSIM↑ | LPIPS↓ | WS-PSNR↑ | SSIM↑ | LPIPS↓ | WS-PSNR↑ | SSIM↑ | LPIPS↓ |
| S-NeRF | 15.25 | 0.579 | 0.546 | 14.16 | 0.563 | 0.580 | 13.13 | 0.523 | 0.607 |
| OmniSyn | 22.90 | 0.850 | 0.244 | 20.31 | 0.790 | 0.317 | 18.91 | 0.761 | 0.354 |
| IBRNet* | 25.72 | 0.855 | 0.258 | 21.69 | 0.751 | 0.382 | 20.04 | 0.706 | 0.431 |
| NeuRay* | 24.92 | 0.832 | 0.260 | 21.92 | 0.766 | 0.347 | 19.85 | 0.715 | 0.407 |
| PanoGRF | **27.12** | **0.876** | **0.195** | **23.38** | **0.811** | **0.282** | **20.96** | **0.761** | **0.352** |

\* means models are trained with cubemap projection and evaluated with equirectangular projection.

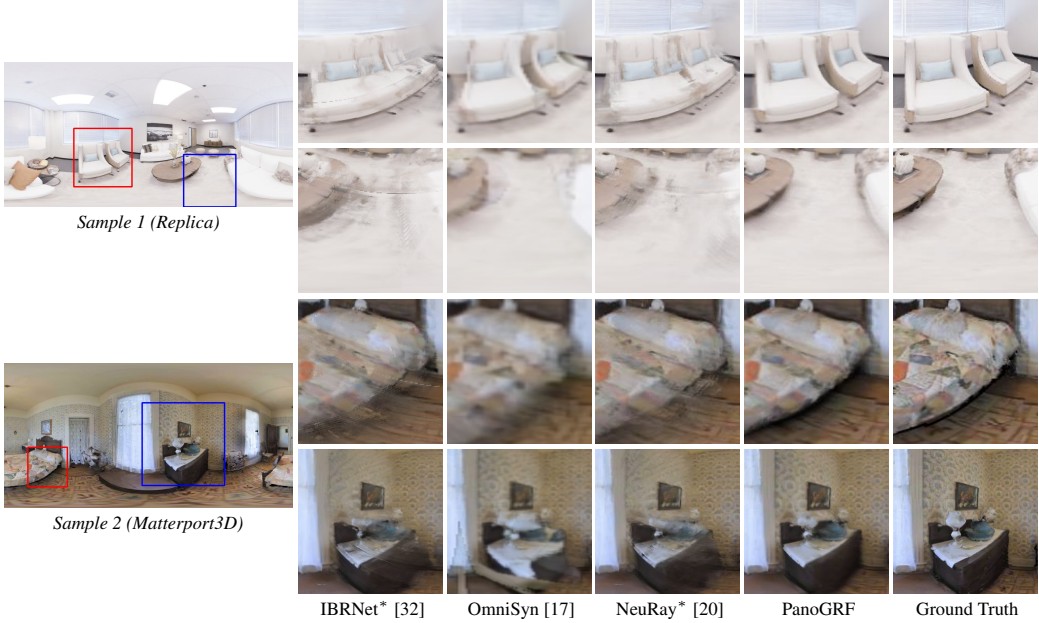

*Sample 1 (Replica)*

*Sample 2 (Matterport3D)*

IBRNet* [32]    OmniSyn [17]    NeuRay* [20]    PanoGRF    Ground Truth

Figure 3: Qualitative comparison on Replica and Matterport3D between IBRNet*, OmniSyn, NeuRay* and PanoGRF.

**Analysis** On Matterport3D and Replica, we quantitatively compared PanoGRF with the baseline methods. The results can be found in Table. 1 and Table. 2. Almost all the metrics show that PanoGRF outperforms other methods significantly. We show the qualitative comparisons in Fig. 3. IBRNet* and NeuRay* aggregate features based on perspective projection. Due to the limited field of view, 3D sample points are often projected somewhere behind the source perspective camera ($z$-depth<0) or outside the image border during pixel alignment. In this case, the aggregated features are incorrect, causing bad rendering results. As for OmniSyn, its rendering outputs exhibit ghosting artifacts, particularly notable at the boundaries of the two sofas in Sample 1 of Fig. 3. Unlike these methods, PanoGRF is a generalizable spherical NeRF which is more appropriate for panoramic images due to the use of spherical coordinates. It achieves superior accuracy in synthesizing object boundaries and demonstrates better pixel alignment, as evident from the results showcasing the sofas in Sample 1 and the desk in Sample 2 of Fig. 3). On Residential, we also compared PanoGRF with S-NeRF, IBRNet*, NeuRay*. The results of S-NeRF show severe floaters (See the ceilings of Sample 1 in Fig. 4). As

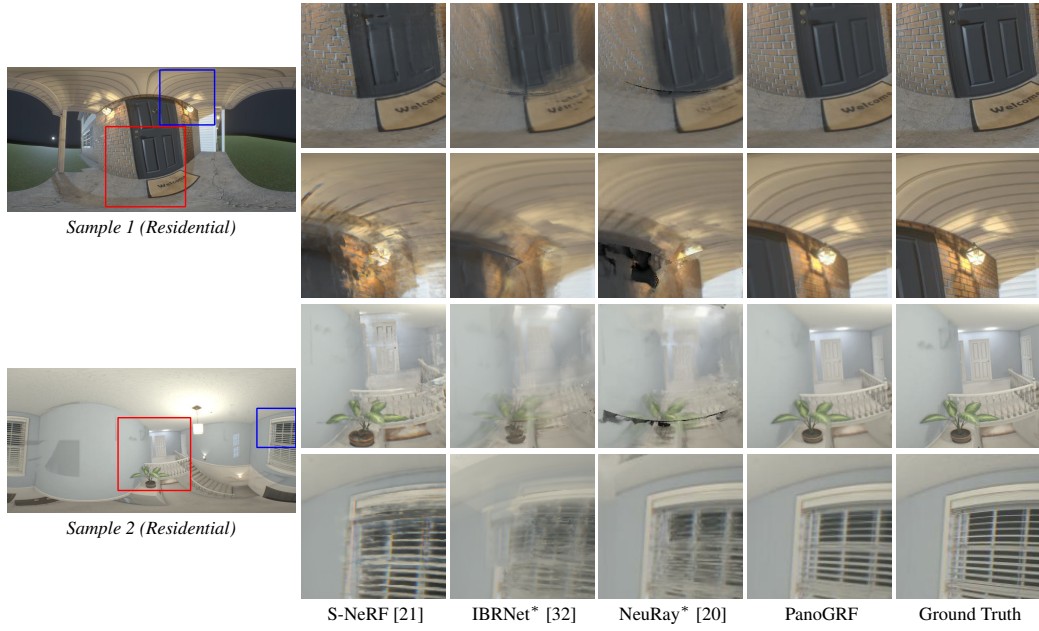

Sample 1 (Residential)

Sample 2 (Residential)

| S-NeRF [21] | IBRNet* [32] | NeuRay* [20] | PanoGRF | Ground Truth |

Figure 4: Qualitative comparison on Residential between S-NeRF, IBRNet*, NeuRay* and PanoGRF.

converting panoramas into perspective views brings information loss, the perspective generalization methods IBRNet* and NeuRay* have notable artifacts. In contrast, PanoGRF demonstrates excellent generalization performance on the Residential dataset, as shown in Fig. 4. Additional qualitative comparisons can be seen in the supplementary material.

Table 2: Quantitative comparison with baseline methods on Replica and Residential Dataset

| Dataset | Replica (1.0m) | | | | Residential (about 0.3m) | | | |
|---|---|---|---|---|---|---|---|---|
| method | PSNR↑ | WS-PSNR↑ | SSIM↑ | LPIPS↓ | PSNR↑ | WS-PSNR↑ | SSIM↑ | LPIPS↓ |
| S-NeRF | 16.91 | 16.10 | 0.723 | 0.443 | 23.06 | 22.47 | 0.741 | 0.435 |
| OmniSyn | 23.91 | 23.17 | 0.898 | 0.189 | - | - | - | - |
| IBRNet* | 23.35 | 22.65 | 0.854 | 0.291 | 22.95 | 22.47 | 0.735 | 0.498 |
| NeuRay* | 26.62 | 25.90 | 0.899 | 0.187 | 23.01 | 22.38 | 0.753 | 0.427 |
| PanoGRF | **30.08** | **29.22** | **0.937** | **0.134** | **31.64** | **31.03** | **0.909** | **0.207** |

## 4.4 Ablation Studies

We conducted ablation studies to evaluate the effectiveness of the key components in mono-guided spherical depth estimator: $360°$ monocular depth and multi-view stereo. We provide qualitative comparisons of $360°$ depth estimation quality on Matterport3D in Fig. 5, and analyze the view synthesis quality on Replica in Table. 3. More results can be found in the supplementary material.

$360°$ **monocular depth**  We removed the $360°$ monocular depth guidance and kept the total number of depth candidates unchanged ($N_{uni} = 64$) in the depth sampling. Under three camera baselines, all the metrics dropped after removing the $360°$ monocular depth. Especially, under the camera baseline of 2.0 meters, WS-PSNR dropped by 1.33dB without monocular depth. This removal resulted in error-prone object boundaries, such as the bedside and wooden pillar (shown in Fig. 6). The guidance of monocular depth can give more accurate depth candidates and mitigates the view inconsistency problem, which $360°$ multi-view stereo cannot handle alone.

$360°$ **multi-view stereo**  We removed $360°$ multi-view stereo and used $360°$ monocular depth directly as the input of the geometry feature extraction in PanoGRF. In this case, WS-PSNR dropped

more than 1.0 dB under the camera baselines of 1.5 and 2.0 meters. And LPIPS became 0.165 from 0.134 under the baseline of 1.0 meters. Without multi-view stereo, multi-view consistency of geometry is not guaranteed, degrading the quality of novel view synthesis results. In Fig. 6, we observed the presence of ghosting artifacts near the left wooden pillar when relying solely on 360° monocular depth.

Table 3: Ablation studies on Replica

| baseline | 1.0m | | | 1.5m | | | 2.0m | | |
|---|---|---|---|---|---|---|---|---|---|
| method | WS-PSNR↑ | SSIM↑ | LPIPS↓ | WS-PSNR↑ | SSIM↑ | LPIPS↓ | WS-PSNR↑ | SSIM↑ | LPIPS↓ |
| w/o Mono | 28.71 | 0.935 | 0.136 | 25.53 | 0.898 | 0.188 | 23.33 | 0.864 | 0.249 |
| w/o MVS | 28.23 | 0.925 | 0.165 | 25.15 | 0.888 | 0.227 | 23.15 | 0.855 | 0.274 |
| full | **29.22** | **0.937** | **0.134** | **26.38** | **0.903** | **0.187** | **24.48** | **0.885** | **0.223** |

# 5 Conclusion

In this paper, we propose PanoGRF, generalizable spherical radiance fields for wide-baseline panoramas. We aggregate the appearance and geometry features from input panoramas through spherical projection to avoid panorama-to-perspective conversion. To address the view inconsistency problem, we use 360° monocular depth to guide the spherical depth sampling and obtain a more accurate geometric prior for the spherical radiance fields. Experiments on multiple datasets verify that PanoGRF can render high-quality novel views given wide-baseline panoramas.

**Limitations** Similar to other generalizable radiance fields, PanoGRF suffers from the issue of rendering speed. Additionally, since we only train PanoGRF on indoor data due to the lack of large-scale outdoor 360° datasets, its generalization performance can be limited when applied to outdoor scenes with significantly different depth scales.

**Societal impact** This method may be used to synthesize fake or deceptive panoramas, combined with generative methods.

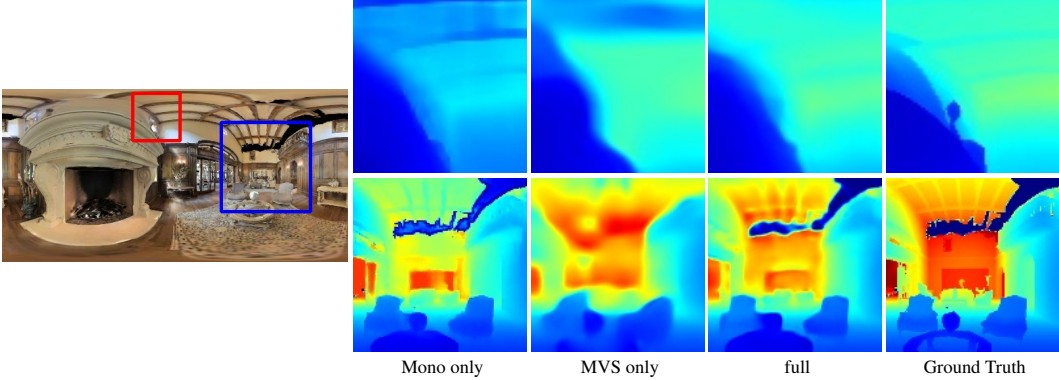

Figure 5: Qualitative results of ablation studies for depth estimation on Matterport3D. *Mono only* and *MVS only* respectively refer to the results obtained when using only 360° monocular depth and 360° multi-view stereo. The use of 360° monocular depth alone does not ensure multi-view consistency, resulting in potential discrepancies between the predicted scale and the ground truth in certain regions. Due to the occlusion problem, using only 360° MVS results in inaccurate and less detailed depth predictions, especially at the boundaries of objects.

# 6 Acknowledgements

This work was supported by the Natural Science Foundation of China (Project Number 62132012) and Tsinghua-Tencent Joint Laboratory for Internet Innovation Technology.

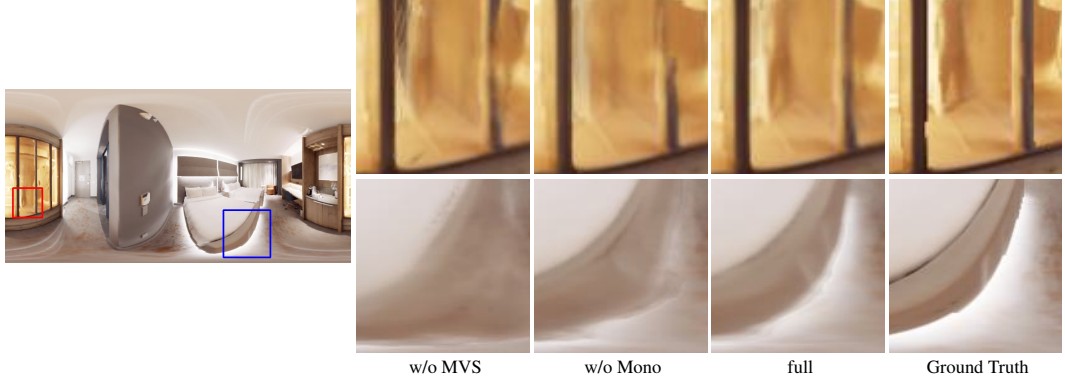

| w/o MVS | w/o Mono | full | Ground Truth |

Figure 6: Qualitative results of ablation studies for novel view synthesis on Replica.

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

Table 4: Additional ablation studies on Matterport3D

| Dataset | Matterport3D (1.0m) | | | |
|---|---|---|---|---|
| method | PSNR↑ | WS-PSNR↑ | SSIM↑ | LPIPS↓ |
| w/o appearance | 5.44 | 5.24 | 0.001 | 0.707 |
| w/o geometry | 26.25 | 25.25 | 0.839 | 0.263 |
| full | **28.10** | **27.10** | **0.876** | **0.195** |

# A    More Results of Qualitative Comparisons

Qualitative comparisons with baseline methods on Replica and Matterport3D can be seen in Fig. 7 and Fig. 8, respectively.

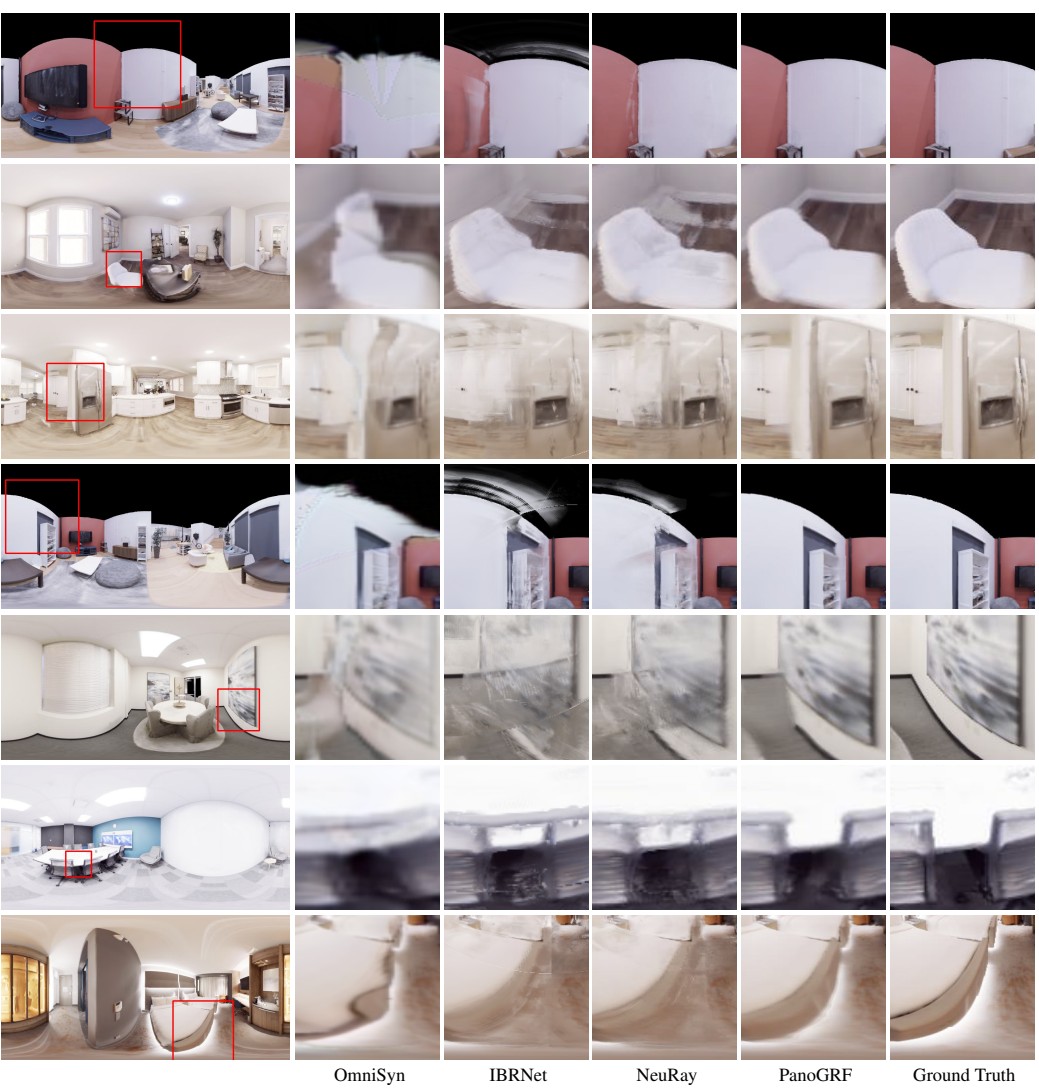

OmniSyn          IBRNet          NeuRay          PanoGRF          Ground Truth

Figure 7: Qualitative comparisons with baseline methods on Replica.

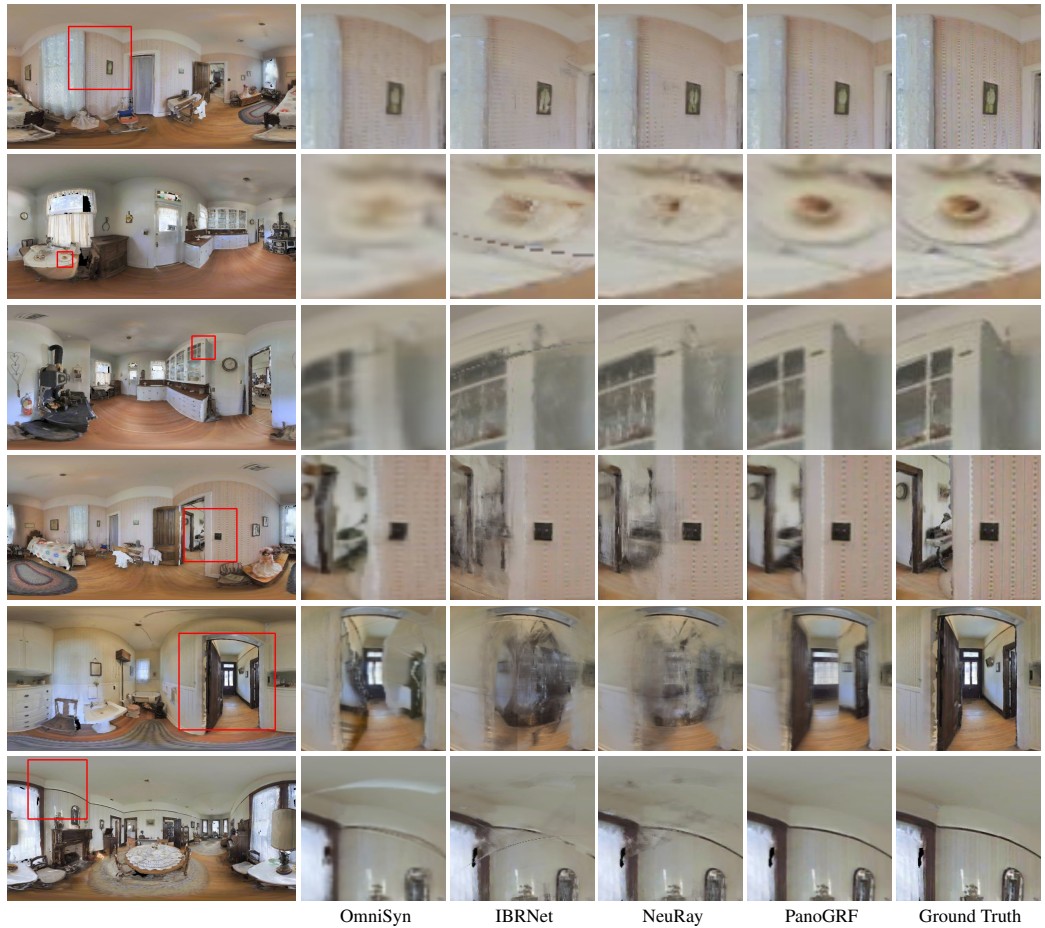

Figure 8: Qualitative comparisons with baseline methods on Matterport3D.

Table 5: Ablation studies for depth estimation on Matterport3D

| setting | $L_1\downarrow$ | $L_2\downarrow$ | RMSE$\downarrow$ | WS-$L_1\downarrow$ | WS-$L_2\downarrow$ | WS-RMSE$\downarrow$ |
|---|---|---|---|---|---|---|
| MVS only | 0.1731 | 0.5048 | 0.5831 | 0.1984 | 0.2806 | 0.4731 |
| Mono only | 0.2452 | 0.3175 | 0.4731 | 0.2445 | 0.2729 | 0.4522 |
| full | **0.1441** | **0.2047** | **0.3877** | **0.1502** | **0.1624** | **0.3546** |

Table 6: The impact of different backbones for depth estimation on Matterport3D

| setting(backbone) | $L_1\downarrow$ | $L_2\downarrow$ | RMSE$\downarrow$ | WS-$L_1\downarrow$ | WS-$L_2\downarrow$ | WS-RMSE$\downarrow$ | parameters(M) |
|---|---|---|---|---|---|---|---|
| MVS(resnet-18) | 0.1731 | 0.5048 | 0.5831 | 0.1984 | 0.2806 | 0.4731 | 29.75 |
| MVS(resnet-34) | 0.1654 | 0.4820 | 0.5676 | 0.1844 | 0.2577 | 0.4493 | 39.39 |
| MVS(resnet-50) | 0.1598 | 0.4725 | 0.5630 | 0.1822 | 0.2446 | 0.4442 | 47.10 |
| MVS(resnet-101) | 0.1642 | 0.4994 | 0.5717 | 0.1835 | 0.2519 | 0.4441 | 65.22 |
| MVS(resnet-18)+Mono | **0.1441** | **0.2047** | **0.3877** | **0.1502** | **0.1624** | **0.3546** | 58.95 |
| MVS(resnet-34)+Mono | 0.1549 | 0.2231 | 0.4120 | 0.1684 | 0.1878 | 0.3844 | 68.59 |
| MVS(resnet-50)+Mono | 0.1519 | 0.2379 | 0.4188 | 0.1675 | 0.1955 | 0.3887 | 76.30 |
| MVS(resnet-101)+Mono | 0.1735 | 0.2673 | 0.4452 | 0.1831 | 0.2129 | 0.4023 | 94.42 |

Table 7: The impact of different numbers of depth candidates $N_{mono}$ for depth estimation on Matterport3D

| $N_{mono}$ | $L_1 \downarrow$ | $L_2 \downarrow$ | RMSE$\downarrow$ | WS-$L_1\downarrow$ | WS-$L_2\downarrow$ | WS-RMSE$\downarrow$ |
|---|---|---|---|---|---|---|
| 1 | 0.1586 | 0.2498 | 0.4317 | 0.1745 | 0.1971 | 0.3937 |
| 3 | **0.1432** | **0.1993** | **0.3865** | 0.1529 | 0.1649 | 0.3580 |
| 5 | 0.1441 | 0.2047 | 0.3877 | **0.1502** | **0.1624** | **0.3546** |
| 7 | 0.1496 | 0.2252 | 0.4104 | 0.1640 | 0.1896 | 0.3832 |
| 9 | 0.1645 | 0.2912 | 0.4511 | 0.1752 | 0.2215 | 0.4059 |
| 16 | 0.1596 | 0.2379 | 0.4188 | 0.1675 | 0.1955 | 0.3887 |
| 32 | 0.1735 | 0.2673 | 0.4452 | 0.1831 | 0.2129 | 0.4023 |
| 48 | 0.1689 | 0.2618 | 0.4333 | 0.1776 | 0.2047 | 0.3941 |
| 64 | 0.1604 | 0.2432 | 0.4162 | 0.1669 | 0.2004 | 0.3853 |

Table 8: The impact of different values of $\sigma$ for depth estimation on Matterport3D

| $\sigma$ | $L_1 \downarrow$ | $L_2 \downarrow$ | RMSE$\downarrow$ | WS-$L_1\downarrow$ | WS-$L_2\downarrow$ | WS-RMSE$\downarrow$ |
|---|---|---|---|---|---|---|
| 0.1 | 0.1544 | 0.2515 | 0.4261 | 0.1672 | 0.1915 | 0.3830 |
| 0.5 | 0.1441 | **0.2047** | **0.3877** | **0.1502** | **0.1624** | **0.3546** |
| 1.0 | 0.1689 | 0.2686 | 0.4424 | 0.1803 | 0.2124 | 0.4029 |
| 1.5 | **0.1426** | 0.2457 | 0.4254 | 0.1563 | 0.1759 | 0.3723 |

## B  Spherical Projection

**Equirectangular-to-spherical**  The transformation from the equirectangular image coordinate system to the polar coordinate system is defined as:

$$\begin{aligned} \phi &= v/H * \pi \\ \theta &= u/W * 2\pi - 0.5\pi, \end{aligned} \tag{6}$$

where $\phi$, $\theta$ represent the latitude and longitude of the sphere, $u$, $v$ represent the rows and columns of the panorama, $H$ and $W$ represent the height and width of the panorama respectively.

**Spherical-to-cartesian**  The transformation from the polar coordinate system to the 3D Cartesian coordinate system is:

$$\begin{aligned} x &= sin(\phi) * cos(\theta) \\ y &= cos(\phi) \\ z &= sin(\phi) * sin(\theta). \end{aligned} \tag{7}$$

**Cartesian-to-spherical**  The camera cartesian coordinate $(x, y, z)$ is transformed into the polar coordinate $(\theta, \phi, t)$ ($t \in \mathbb{R}^+$ denotes its spherical depth in view $I_j$) by:

$$\begin{aligned} t &= \sqrt{x^2 + y^2 + z^2} \\ \theta &= arctan(\frac{z}{x}) \\ \phi &= arccos(\frac{y}{t}). \end{aligned} \tag{8}$$

Table 9: Quantitative comparison with Cross Attention Renderer [7] on Matterport3D and Replica

| Dataset | Matterport3D (1.0m) | | | Replica(1.0m) | | |
|---|---|---|---|---|---|---|
| method | PSNR↑ | SSIM↑ | LPIPS↓ | PSNR↑ | SSIM↑ | LPIPS↓ |
| CAR [7] | 22.87 | 0.7679 | 0.3108 | 23.60 | 0.8594 | 0.2515 |
| PanoGRF | **27.78** | **0.8158** | **0.2444** | **29.87** | **0.9046** | **0.1604** |

Table 10: Comparisons with NeuRay [20] given multi-view inputs on Matterport3D

| Dataset | Matterport3D (1.0m) | | | |
|---|---|---|---|---|
| method | PSNR↑ | WS-PSNR↑ | SSIM↑ | LPIPS↓ |
| NeuRay [20] | 27.82 | 26.74 | 0.8614 | 0.2312 |
| PanoGRF | **28.99** | **27.91** | **0.8762** | **0.2071** |

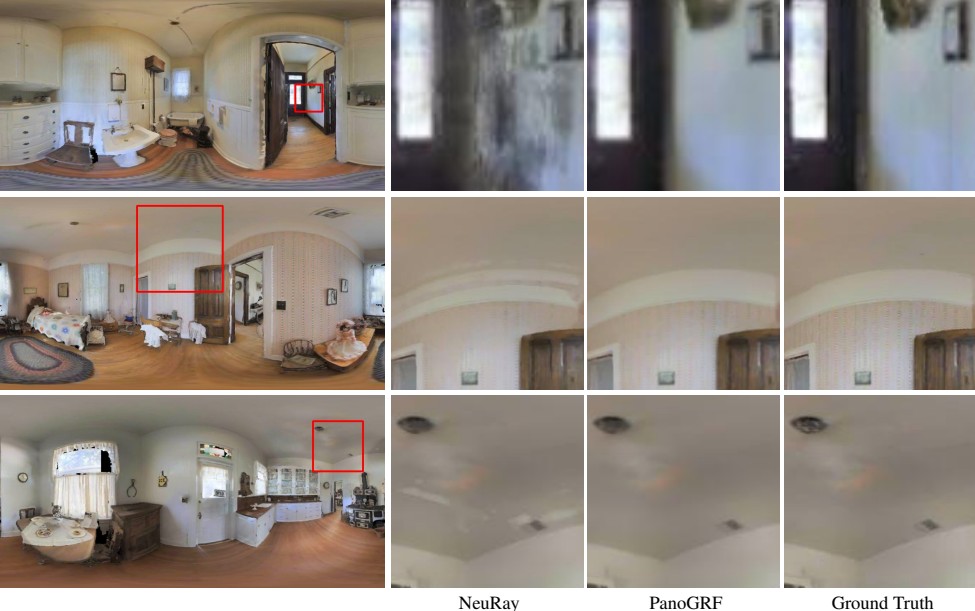

NeuRay        PanoGRF        Ground Truth

Figure 9: Qualitative comparisons between PanoGRF and NeuRay on Matterport3D with multi-view panoramic inputs.

**Spherical-to-equirectangular**    The spherical polar coordinate $(\theta, \phi, t)$ is turned into the equirectangular image coordinate $(u, v)$ by the inverse process of Eq. 6.

## C   More Ablation Studies for $360°$ View Synthesis

We conducted ablation studies on Matterport3D, and the results are shown in Table. 4. In the "w/o appearance feature" ablation study, we replaced the appearance feature vector with a zero vector to disable the appearance feature while keeping other modules unchanged. We found that the model without appearance features loses its ability to infer the color of novel views entirely, as the generalizable renderer heavily relies on appearance cues from input views. In the "w/o geometry feature" ablation study, we replaced the geometry feature vector with a zero vector to disable the geometry feature while keeping other modules unchanged. We observed that although the model can still infer normal results, its performance is significantly worse than the original (full) model.

## D   Comparisons with NeuRay [20] Given Multi-view Inputs

Our method is not limited to two panoramas. For instance, when rendering a test view in the renderer module, we use $N$ input panoramas as reference images, and the renderer does not need to be modified. In the $360°$ spherical depth estimator module, for each reference image, we use the other $N - 1$ input panoramas as source images. We average the multiple cost-volumes obtained during $360°$ multi-view matching process between the reference image and each source image. The rest is unchanged. In this way, our method can be applied to the multi-view panoramic inputs.

To further verify the effectiveness of our method, we conducted comparative experiments with NeuRay using multiple panoramic image as inputs. We placed the four input viewpoints at the

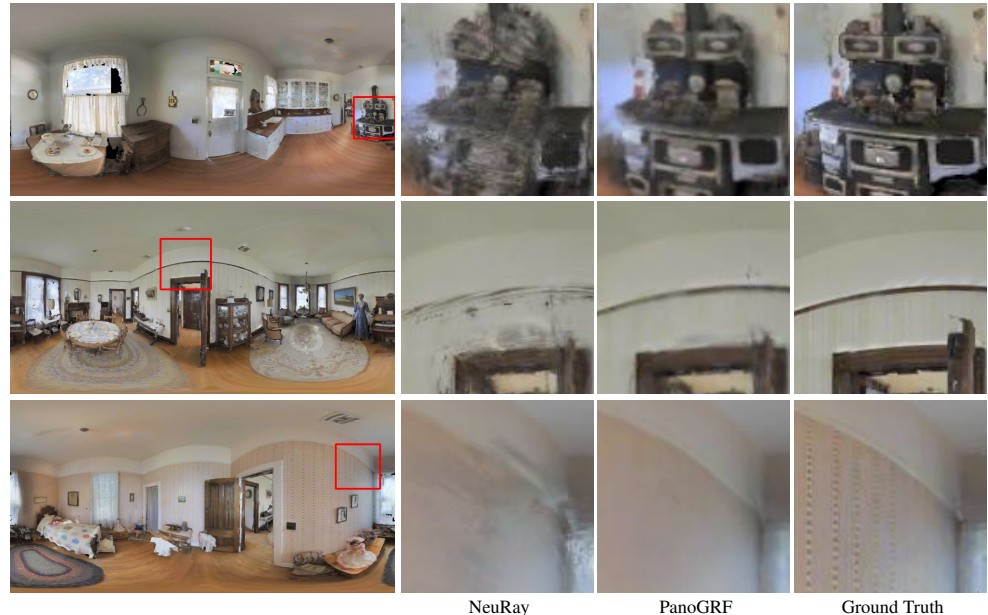

NeuRay       PanoGRF       Ground Truth

Figure 10: Qualitative comparisons with NeuRay beyond the camera baseline on Matterport3D. We synthesized novel views at positions 0.25 meters above the middle point between two input viewpoints. The input viewpoints are 1.0 meters apart. Our method can achieve better results than NeuRay beyond the camera baseline.

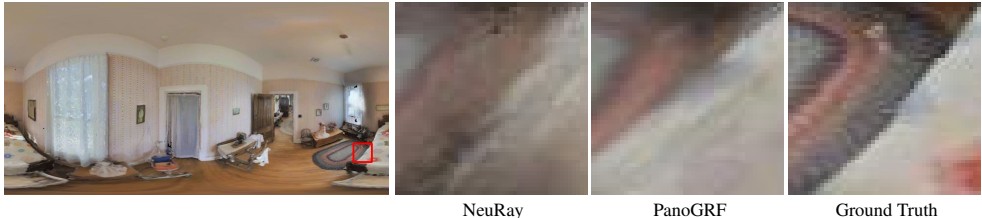

NeuRay       PanoGRF       Ground Truth

Figure 11: Failure case. The carpet area behind the bed was not visible in the input viewpoints, but it becomes visible in higher viewpoints. NeuRay and PanoGRF tend to produce different and blurry results compared to the ground truth because they lack generative power. Combining existing diffusion generative models could potentially address this issue. We leave this as future work.

corners of a horizontal square and tested the rendering performance at the center viewpoint and other viewpoints located at -0.4, -0.2, -0.1, -0.05, 0.05, 0.1, 0.2, and 0.4 meters in the vertical direction from the center viewpoint. The diagonal length of the square is 1.0 meters. Table. 10 and Fig. 9 present the quantitative and qualitative comparison results between PanoGRF and NeuRay. As shown, PanoGRF still largely outperforms NeuRay with multiple panoramic inputs.

This experiment is added during the rebuttal period. Due to the limited time, we trained PanoGRF only for 20k iterations and NeuRay for 80k iterations. The learning decay strategies are similar to the setting of two views.

## E    Comparisons with NeuRay [20] beyond Camera Baseline and Failure Case

We conducted an additional experiment where novel views were generated at positions 0.25 and 0.5 meters above the middle point between two input viewpoints. The input viewpoints are 2.0 meters apart. We compared the quantitative and qualitative results of our method with NeuRay's as shown in Table. 11 and Fig. 10. PanoGRF consistently surpassed NeuRay's performance, indicating its capacity to yield superior results beyond the camera baseline.

We also present the failure case of PanoGRF under such condition in Fig. 11. In some viewpoints, a previously occluded area may be visible and since this area has not been seen in either of the two

Table 11: Comparisons with NeuRay [20] beyond Camera Baseline on Matterport3D

| Dataset | | Matterport3D (1.0m) | | | |
|---|---|---|---|---|---|
| distance | method | PSNR↑ | WS-PSNR↑ | SSIM↑ | LPIPS↓ |
| 0.25m | NeuRay | 20.66 | 20.05 | 0.714 | 0.409 |
| 0.25m | PanoGRF | **21.98** | **21.30** | **0.763** | **0.348** |
| 0.5m | NeuRay | 20.39 | 19.94 | 0.711 | 0.420 |
| 0.5m | PanoGRF | **21.99** | **21.42** | **0.769** | **0.349** |

existing viewpoints, the synthesis of this area is not effective. The area, which has not been seen in either of the existing viewpoints may be able to be filled in by combining with the existing diffusion generative approach. This is the next direction we plan to investigate.

## F  Experiments for Mono-guided Spherical Depth Estimator

### F.1  Ablation Studies for Mono-guided Spherical Depth Estimator

To further validate the effectiveness of the key components, namely $360°$ multi-view stereos and $360°$ monocular depth, we conducted ablation studies specifically focused on spherical depth estimation. For evaluation purposes, we selected three commonly used metrics: $L_1$, $L_2$, and RMSE. Additionally, we also used WS-$L_1$, WS-$L_2$, and WS-RMSE as metrics, which incorporate weighted latitudes of equirectangular images to simulate WS-PSNR [29]. This approach aims to mitigate the impact of equirectangular projection distortion. We selected the first 1000 panorama pairs from Matterport3D as our test data, with a camera baseline of 1 meter. For the evaluation, we considered depth values within the range of $[0.1, 10]$ as valid.

The quantitative results of our ablation studies are presented in Table 5, while the qualitative results can be observed in Fig. 12. The experiments clearly demonstrate the importance of each module in achieving accurate depth estimation. Removing either the $360°$ multi-view stereo or the $360°$ monocular depth significantly reduces the depth accuracy. Using only $360°$ monocular depth does not guarantee multi-view consistency, resulting in potential discrepancies between the predicted scale and the ground truth in certain regions. The occlusion problem poses a challenge for using only $360°$ MVS, particularly at the boundaries of objects. Consequently, the depth predictions in these regions are inaccurate and lack detail.

### F.2  Different Backbones for $360°$ Multi-view Stereo

Introducing $360°$ monocular depth to $360°$ multi-view stereo does result in an increase in the number of model parameters. However, it is important to note that the improvement in depth accuracy is not attributed to the increase in parameters. In our experiments, we increased the model size of $360°$ MVSNet by using larger backbones, specifically ResNet [9].

From Table. 6, we found that $L_2$ and RMSE of $360°$ MVSNet(ResNet-101) are still far inferior to those of $360°$ MVSNet(ResNet-18) together with the $360°$ monocular depth network [13]. This observation suggests that simply increasing the model size does not effectively address the view inconsistency problem in the wide-baseline setting. On the other hand, the introduction of $360°$ monocular depth provides a qualitative improvement to the accuracy of $360°$ multi-view stereo. By incorporating monocular depth information, the model gains additional cues that help mitigate the view inconsistency issue and improve depth estimation performance. Furthermore, when larger backbones were used as replacements, it was found that the performances of $360°$ MVS+Mono deteriorated. This could be attributed to the excessive model parameters leading to overfitting.

### F.3  Different Hyperparameters for Mono-guided Spherical Depth Estimator

We conducted evaluations to assess the impact of different values for $\sigma$ (standard deviation) and $N_{mono}$ (number of monocular depth candidates) on mono-guided spherical depth sampling. The results are presented in Table 7 and Table 8.

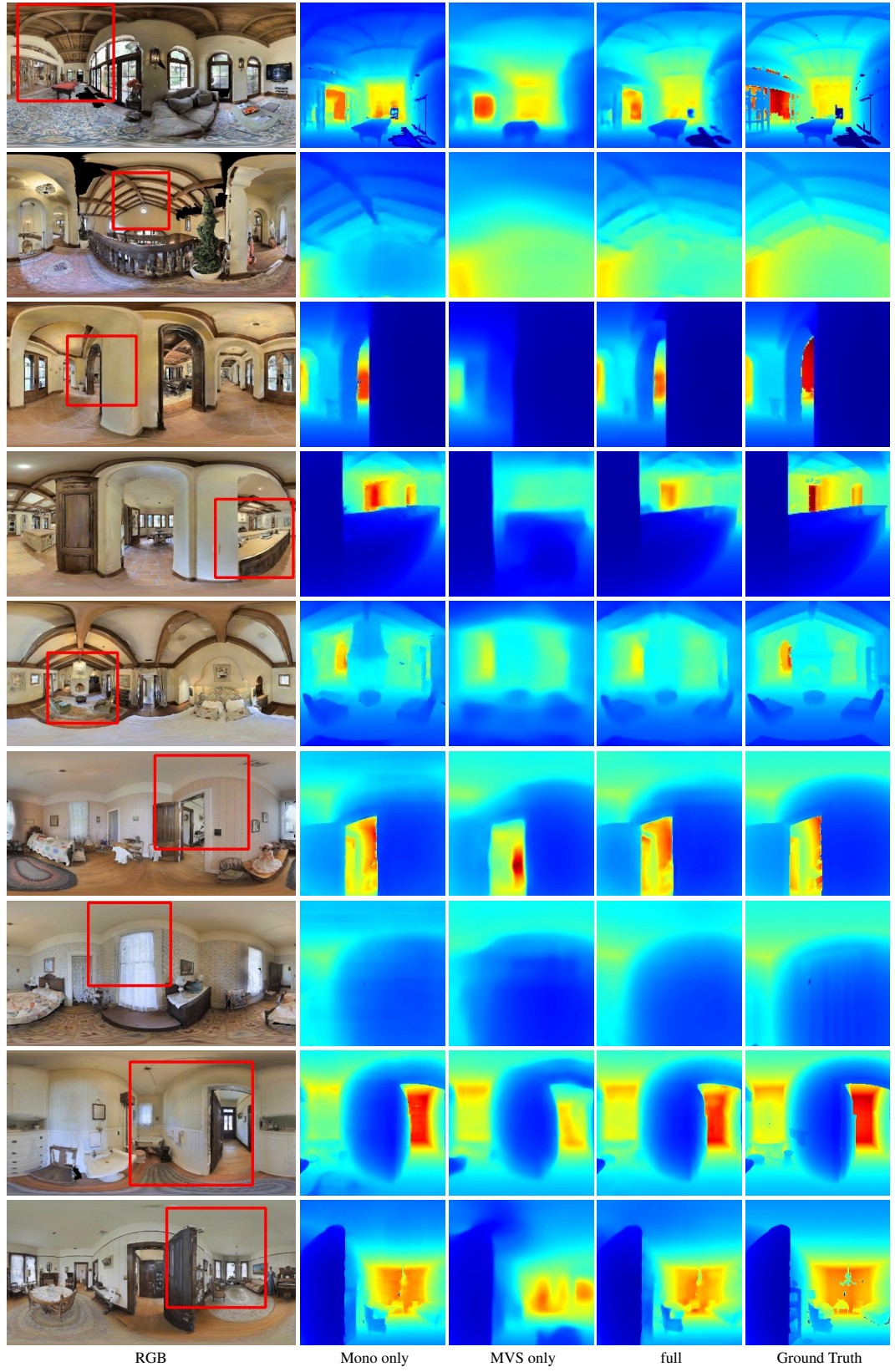

| RGB | Mono only | MVS only | full | Ground Truth |

Figure 12: Qualitative results of ablation studies for depth estimation on Matterport3D. *Mono only* and *MVS only* respectively refer to the results obtained when using only 360° monocular depth and 360° multi-view stereo.

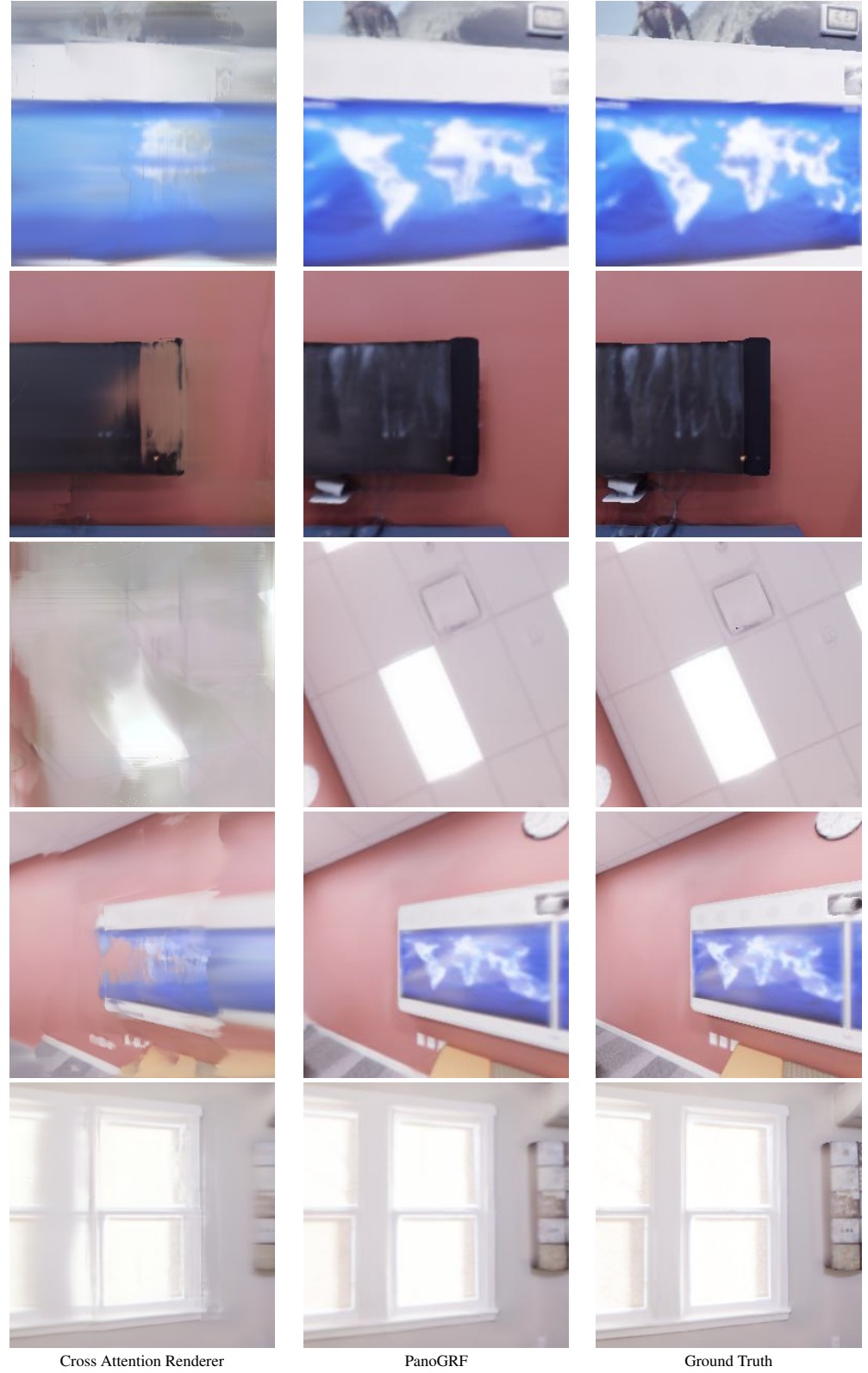

Cross Attention Renderer       PanoGRF       Ground Truth

Figure 13: Qualitative comparisons with Cross Attention Renderer on Replica

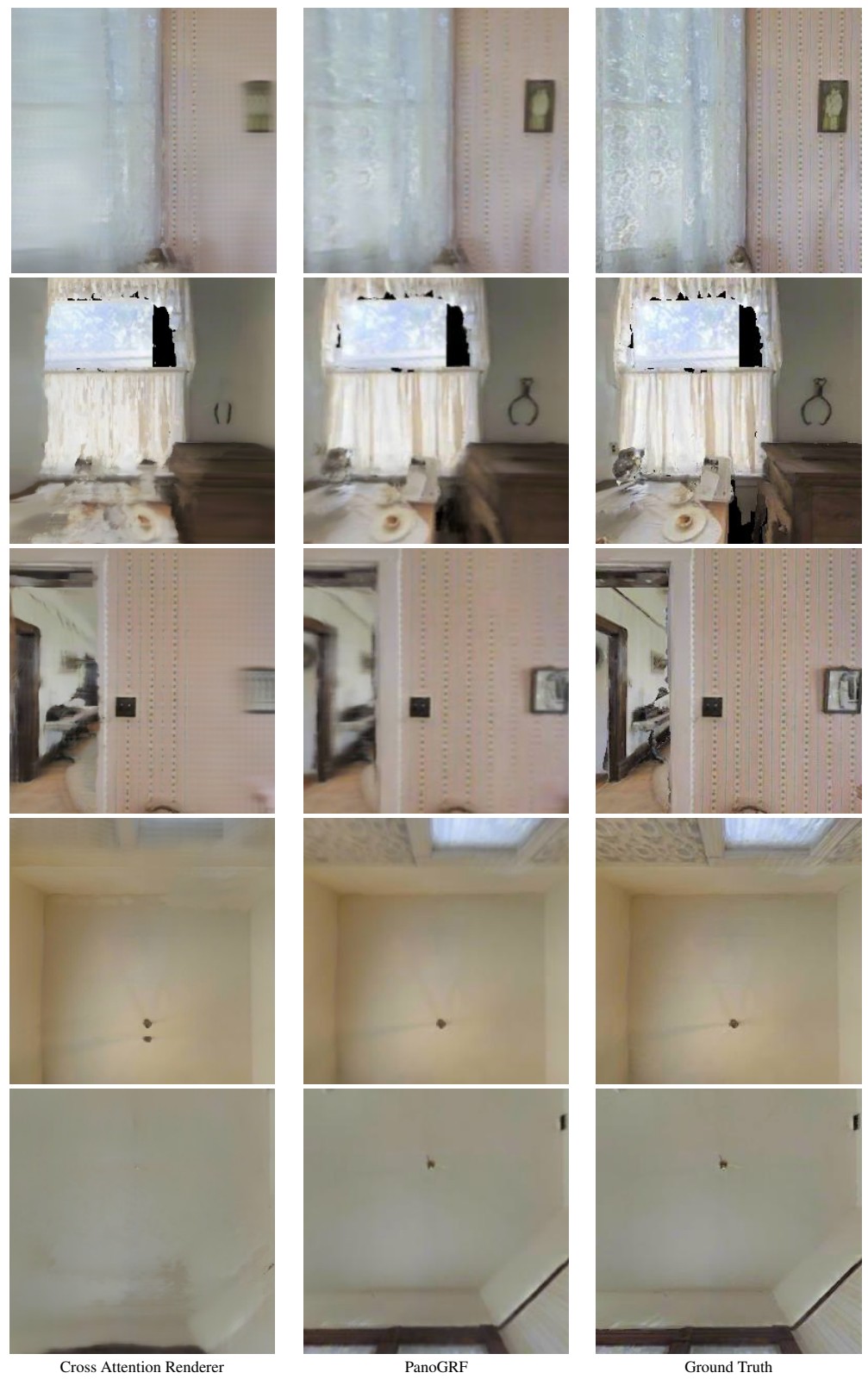

Cross Attention Renderer          PanoGRF          Ground Truth

Figure 14: Qualitative comparisons with Cross Attention Renderer on Matterport3D

Table 12: The quantitative results of fine-tuning of the general renderers. The best results are in bold.

| baseline | | 1.0m | | | | 0.2m | | | |
|---|---|---|---|---|---|---|---|---|---|
| method | setting | PSNR↑ | WS-PSNR↑ | SSIM↑ | LPIPS↓ | PSNR↑ | WS-PSNR↑ | SSIM↑ | LPIPS↓ |
| NeuRay | gen | 27.751 | 27.253 | 0.8470 | 0.2565 | 34.318 | 33.376 | 0.9409 | 0.1158 |
| PanoGRF | gen | **28.818** | **28.487** | **0.8778** | **0.1996** | 35.817 | 35.056 | 0.9554 | 0.0959 |
| NeuRay(+Consist) | ft | 26.755 | 26.250 | 0.8181 | 0.2820 | 33.354 | 32.359 | 0.9240 | 0.1304 |
| PanoGRF | ft | 24.341 | 23.941 | 0.8175 | 0.2834 | **35.828** | **35.138** | **0.9578** | **0.0912** |
| PanoGRF (+Depth) | ft | 27.399 | 27.202 | 0.8556 | 0.2446 | 33.691 | 33.066 | 0.9333 | 0.1342 |

The reliability of 360° monocular depth estimation is not perfect. Therefore, our paper employs uniform sampling to compensate for the remaining depth candidates. In the ablation experiments, we consistently maintain $N_{mono} + N_{uni} = 64$, where $N_{uni}$ represents the sample number of uniform distribution. We discovered that the configuration yields the best results with regard to the metrics of WS-L1, WS-L2, and WS-RMSE.

## G   Comparisons with Cross Attention Renderer [7]

The Cross Attention Renderer (CAR) [3] is a method that operates on a wide-baseline perspective pair. We divided the two panoramas into cube maps and utilized the corresponding sides of the cube maps as inputs for CAR. Specifically, we rendered the corresponding side of the cube maps at the middle viewpoint. For example, we input the left side of the cube maps and generate the left side of the cube maps at the intermediate viewpoint as the output. We repeated this process for all six corresponding pairs of cube maps. In contrast, our method directly takes two panoramas as input and performs ray casting based on perspective projection. We then render the results for each side of the cube maps at the intermediate viewpoint, preserving the panoramic nature of the input.

We conducted quantitative comparative experiments on Matterport3D and Replica. The qualitative comparisons with CAR on Replica and Matterport3D can be seen in Fig. 13 and Fig. 14. The results clearly demonstrate that PanoGRF significantly outperforms CAR in terms of rendering quality. CAR suffers from limitations associated with its input field-of-view (FoV), particularly in edge regions that are only visible from a single perspective view. CAR relies on a pure stereo-matching method for geometric estimation, which leads to suboptimal rendering performance, as evidenced by the results presented in Table. 9. In contrast, PanoGRF is specifically designed to handle full FoV inputs, and it mitigates the issue of view inconsistency by incorporating the 360° monocular depth network.

## H   Fine-tuning of PanoGRF

We conducted per-scene fine-tuning for NeuRay [20] and PanoGRF on the first test scene of Matterport3D with baselines of 1.0 and 0.2 meters, respectively. The general renderers were fine-tuned for 10k iterations, and the quantitative results are presented in Table.12. Under the baseline of 1.0 meters, the general renderer of PanoGRF, denoted as PanoGRF-gen, achieved the best performance. NeuRay-ft, the fine-tuned renderer of NeuRay, underwent fine-tuning using RGB loss and depth consistency loss, following the methodology described in their original paper [20]. PanoGRF-ft was fine-tuned with only RGB loss. However, the results of NeuRay-ft and PanoGRF-ft were inferior to NeuRay-gen and PanoGRF-gen, respectively. This suggests that fine-tuning under a wide-baseline setting does not improve the performance of general renderers. It is likely that fine-tuning with a wide baseline leads to overfitting. Even with the introduction of a depth uncertainty loss [24] by supervising the renderer depth of PanoGRF with predicted spherical depths during the fine-tuning process, the results of PanoGRF-ft remained inferior to those of the general renderer of PanoGRF. On the other hand, the performance of PanoGRF-ft under the baseline of 0.2 meters was slightly better than that of PanoGRF-gen. However, adding the depth loss [24] was still unable to improve PanoGRF-gen when fine-tuning. Inaccurate predicted depths in certain regions may be the cause, misleading the estimation of NeRF's geometry and thereby reducing the rendering performance.

---

[3]Cross Attention Renderer: https://github.com/yilundu/cross_attention_renderer

Additionally, NeuRay-ft was still inferior to NeuRay-gen under the narrow baseline. This may be attributed to the limited field-of-view, which can result in the aggregation of incorrect features. When projecting 3D sample points onto other source perspective views (cube-maps), these points may fall outside the image borders of the source perspective view or be located behind the source perspective camera.

## I  Quantitative Comparisons with Baseline Methods for Narrow-baseline Panoramas

Table 13: Quantitative comparisons with baseline methods on Matterport3D under the baseline of 0.2 and 0.5 meters. The best results are in bold.

| baseline | 0.2m | | | | 0.5m | | | |
|---|---|---|---|---|---|---|---|---|
| method | PSNR↑ | WS-PSNR↑ | SSIM↑ | LPIPS↓ | PSNR↑ | WS-PSNR↑ | SSIM↑ | LPIPS↓ |
| S-NeRF | 20.79 | 19.52 | 0.6967 | 0.3756 | 17.95 | 16.81 | 0.6278 | 0.4856 |
| OmniSyn | 28.95 | 28.26 | 0.9132 | 0.1804 | 26.59 | 26.07 | 0.8897 | 0.2005 |
| IBRNet | 30.53 | 29.63 | 0.9271 | 0.1363 | 28.22 | 27.26 | 0.8844 | 0.1987 |
| NeuRay | 33.54 | 32.33 | 0.9485 | 0.1074 | 30.88 | 29.81 | 0.9196 | 0.1536 |
| PanoGRF | **34.29** | **33.27** | **0.9515** | **0.0977** | **31.41** | **30.46** | **0.9238** | **0.1318** |

We compared PanoGRF with baseline methods under the baseline of 0.2 and 0.5 meters on Matterport3D. As shown in Table. 13, the quantitative results demonstrate that PanoGRF consistently outperforms all the baseline methods under the baseline of 0.2 and 0.5 meters. These findings indicate that our method is applicable to both wide-baseline and narrow-baseline panoramas. In comparison to generalizable methods designed for perspective views, our method is particularly well-suited for synthesizing panoramic views by leveraging the aggregated features based on spherical projection.

## J  More Details of PanoGRF

### J.1  Renderer

#### J.1.1  Training

PanoGRF employs the Adam optimizer [16] with an initial learning rate of 4.0e-4. The pre-training process of PanoGRF was conducted on an A100 GPU for 100k iterations, which required approximately two days. The learning rate is halved every 20k iterations, and a batch size of 512 was used during training.

To streamline the experiments, we train PanoGRF and all the baselines under a 1.0m baseline for comparisons and test them under various baselines. In the ablation studies, the spherical depth estimators (Mono, MVS, and fusion) are pre-trained under a fixed 1.0m baseline. The general renderers in the ablation studies are pre-trained under corresponding baselines when tested under various baselines. We discovered that if we pre-trained the general renderers under a 1.0m baseline and tested them under larger baselines, there was no noticeable difference in performance between them.

#### J.1.2  Architecture

We adopted a coarse-to-fine sampling approach, similar to NeRF [21], and sampled 64 points in both phases. We followed a similar architecture as NeuRay [20] to build our renderers. The coarse and fine renderers share the same image encoder, geometry feature extractor, and visibility encoder. But they have different distribution decoders and aggregation networks $\mathcal{F}$, similar to NeuRay. During training, the weights of the $360°$ spherical depth estimator are fixed due to GPU memory limitations. Our image encoder, visibility encoder, distribution decoder, and aggregation network are implemented similarly to NeuRay, except for the padding mode. In the convolution layers of the image encoder and visibility encoder, we employ circular padding horizontally and zero padding vertically instead of direct zero padding. This is done to adapt to equirectangular image inputs and simulate Circular

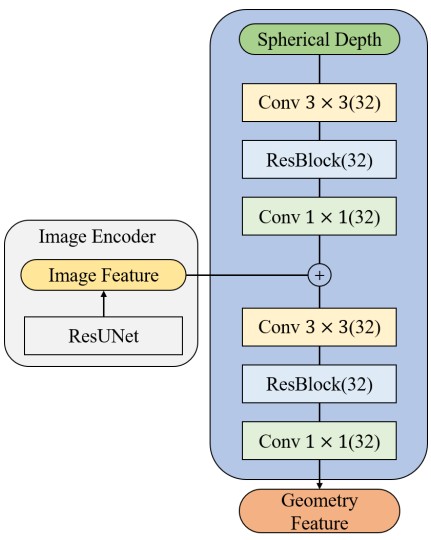

Figure 15: Architecture of geometry feature extractor.

CNNs [27]. The circular padding helps account for the continuity of pixels at the leftmost and rightmost edges of equirectangular images, which are neighbors, while the top and bottom edges are not. The distribution network consists of 5 sub-networks, each with 3 fully connected layers. The appearance feature extractor is a ResNet [9] which contains 13 residual blocks and outputs the appearance feature map with 32 channels. The architecture details of the geometry feature extractor are shown in Fig. 15. The image encoder in the geometry feature extractor contains 14 residual blocks. The spherical depth input for the geometry feature extractor is first downsampled to 1/4 of its original resolution and then fed into the extractor to reduce GPU occupancy.

## J.2    Spherical Depth Estimator

### J.2.1    Training

We initially train the $360°$ monocular depth network and then freeze its weights when training the remaining components of $360°$ MVSNet. The Adam optimizer is employed with a fixed learning rate of 0.0001. Both the $360°$ monocular depth network and $360°$ MVSNet are trained for 100k iterations, which required approximately one day on a single V100 GPU. The batch size is set to 2.

### J.2.2    Architecture

The $360°$ monocular depth network [13] and $360°$ MVSNet utilize ResNet-18 as the feature extractor. The 3D CNN regularization network is composed of 3 downsampling and 3 upsampling blocks, similar to [17]. The depth decoder consists of 2 convolution blocks. The feature map obtained from the middle layer in the $360°$ monocular depth network is concatenated with the regularized spherical cost volume and then decoded into $360°$ depth by the depth decoder. In the multi-view matching process, we compute the similarity by subtracting the feature vectors.

## K    Datasets

For Matterport3D, we split the training and testing set following SynSin [33]. The first 10 scenes of the test set are used for evaluation. In the case of the Replica dataset, we render a total of 18 scenes for evaluation. Additionally, we utilize the Residential dataset provided in [8], which comprises three scenes. For this dataset, we select the first and last panorama as the input views.

## L  Training Details of Baseline Methods

We trained NeuRay [20] and IBRNet [32] for 400k and 250k iterations, respectively. Spherical NeRF (S-NeRF) [21] underwent training for 2000 epochs, equivalent to approximately 256k iterations with the batch size of 4096. For OmniSyn [17], the in-painting network was trained for 50k iterations, which took approximately 3 days on a TitanRTX GPU. The depth estimator of OmniSyn was trained for 100k iterations. Due to limitations in GPU memory, OmniSyn was trained at a resolution of $256 \times 256$, and its output was resized to $512 \times 1024$ for evaluation purposes. CAR [7] was trained for 300k iterations.

