# OpenReview forum: "PanoGRF: Generalizable Spherical Radiance Fields for Wide-baseline Panoramas"
_NeurIPS.cc/2023/Conference — NeurIPS 2023 poster_

### Official Review · Reviewer_WLFo · 2023-06-20

**Soundness:** 3 good
**Presentation:** 4 excellent
**Contribution:** 3 good
**Rating:** 7
**Confidence:** 5

**Summary:**

They describe a novel method for panoramic view synthesis given wide-baseline panoramas as input.  They first obtain depth maps for the input panoramas use a custom spherical depth estimation method guided by a pre-trained monocular depth estimator.  They then extract geometry and appearance features from the input panoramas.  To render a new panorama, they sample points along the viewing rays,  project them into the feature grids, aggregate features using an estimated visibility term, decode the aggregated features into color+density, and compute final color using volume rendering.  They outperform recent generalizable view synthesis methods on Matterport3D, Replica, and Residential datasets.

**Strengths:**

They combine many ideas from prior work on perspective and spherical view synthesis but synthesizing these ideas into a novel approach.  The closest work is probably NeuRay, which is a generalizable radiance field representation for perspective images.  The main differences to NeuRay are that they work in spherical coordinates and they use monocular depth estimation to contribute to the visibility prediction and depth-guided sampling.

Their evaluation is thorough and includes a comparison to a reasonable selection of recent view synthesis methods, including some designed for panoramas and some adapted to panoramas by applying the method to each side of a cube map.  Quantitatively they show a large improvement in rendering quality and this is also evident from the example images in the figures and the videos in the supplementary.

They also include an ablation study in which they verify the usefulness of including 360 monocular depth and multi-view stereo in the method.

Another contribution is the insight (reported in the supplemental) that fine-tuning did not lead to an improvement in rendering quality in their experiments, contrary to what was reported by NeuRay for example.  Instead fine-tuning led to overfitting.  They hypothesize that this is due to the use of wide-baseline images.  This is an interesting result and points to potential future work to overcome this limitation.

**Weaknesses:**

In many ways the work is an adaptation of NeuRay to the spherical case.  It could be argued that the work is somewhat incremental in that sense.  However, their results for panoramic view synthesis are excellent and there is not a lot of previous work on wide-baseline panoramic view synthesis.

Because the name includes GRF, I would think it appropriate to mention and cite GRF which is an earlier general radiance field approach which is a precursor to NeuRay:

Trevithick, A., & Yang, B. (2021). GRF: Learning a general radiance field for 3d representation and rendering. In Proceedings of the IEEE/CVF International Conference on Computer Vision (pp. 15182-15192).

Another weakness is that the provided video only shows planar movement.

**Questions:**

The videos only demonstrate camera movement along a line with no vertical movement.  Does the method support complete 6DOF motion or is limited to planar motion?

**Limitations:**

The discussion of limitations is rather brief.  They could explore failure cases more and highlight cases where the view synthesis result could be improved.

---

> ### Author Rebuttal · Authors · 2023-08-09
>
>
> Dear Reviewer WLFo:
>
>
> Thank you for the review and comments. We are glad to see your positive comments and score. We hope the following responses could solve your concerns.
> * For Weakness1:
> >In many ways the work is an adaptation of NeuRay to the spherical case. It could be argued that the work is somewhat incremental in that sense. However, their results for panoramic view synthesis are excellent and there is not a lot of previous work on wide-baseline panoramic view synthesis.
>
>     We appreciate your thoughtful comments. Although our method is inspired by using NeuRay to the spherical case, it goes much more beyond that. To address the occlusion issue, we additionally employ 360&deg; monocular depth to guide the spherical depth sampling in sphere sweeps of 360&deg; MVSNet. More insights behind our design can be seen in the global response.
>
> * For Weakness 2:
> >Because the name includes GRF, I would think it appropriate to mention and cite GRF which is an earlier general radiance field approach which is a precursor to NeuRay...
>
>     Thanks for your nice suggestion. We will mention and cite GRF in the related work.
> * For Question:
> >The videos only demonstrate camera movement along a line with no vertical movement. Does the method support complete 6DOF motion or is limited to planar motion?
>
>     Yes, PanoGRF support complete 6DOF motion and is not limited to planar motion. To validate this point,  we conducted an additional experiment where we synthesized novel views at positions 0.25 and 0.5 meters above the middle point between two input viewpoints. The input viewpoints are 1.0 meters apart. We compared the quantitative results of our method with NeuRay's using various indicators, as shown in the table. Our results consistently outperformed NeuRay's, demonstrating that our method can achieve better results than NeuRay beyond the camera baseline. For the qualitative comparison, please refer to the attached PDF of our "global" rebuttal.
>
>     |distance|method|PSNR|WS-PSNR|SSIM|LPIPS|
>     |:----:|:----:|:----:|:----:|:----:|:----:|
>     |0.25m|NeuRay|20.66|20.05|0.714|0.409|
>     |0.25m|PanoGRF|**21.98**|**21.30**|**0.763**|**0.348**|
>     |0.5m|NeuRay|20.39|19.94|0.711|0.420|
>     |0.5m|PanoGRF|**21.99**|**21.42**|**0.769**|**0.349**|
>
> * About limitation:
> >The discussion of limitations is rather brief. They could explore failure cases more and highlight cases where the view synthesis result could be improved.
>
>     Great thanks for your suggestion! The qualitative result of the failure case can be seen in the attached PDF of our "global" rebuttal. In some viewpoints, a previously occluded area may be visible, and since this area has not been seen in either of the two existing viewpoints, the synthesis of this area is not effective. The area, which has not been seen in either of the existing viewpoints, may be able to be filled in by combining with the existing diffusion generative approach. This is the next direction we plan to investigate.
>
> Best,
>
> Authors

---

> > ### Comment · Reviewer_WLFo · 2023-08-16
> >
> > Thanks to the authors for their responses.  I have read over the other reviews and the authors' responses.  They have addressed the concerns well and I still would recommend acceptance for this paper.

---

> > > ### Author Response · Authors · 2023-08-17
> > > **Reply of Authors**
> > >
> > > Dear Reviewer WLFo:
> > >
> > > Great thanks for your positive feedback!
> > >
> > > Sincerely,
> > > Authors

---

### Official Review · Reviewer_74Ux · 2023-06-25

**Soundness:** 2 fair
**Presentation:** 1 poor
**Contribution:** 2 fair
**Rating:** 3
**Confidence:** 5

**Summary:**

This work tackels large-baseline (up to 2 meters) multi-view stereo for panorama images. Extending NeuRay framework, the density and raidiance field are estimated based on the feature extracted from nearby views following panorama projection. To predict stereo depth as geometric feature, this work propose to use 360 monocular depth estimation model to sample more sphere sweeps close to the surface. All the features are extracted or estimated by 2D/3D CNN and can be trained from many scenes for generalization.

**Strengths:**

The quantitative and qualitative results are better than the compared baselines.

**Weaknesses:**

**Adapting from perspective cubemap to equirectangular projection is not a big contribution.**
One of the main claimed disadvantage of some of the baseline methods (i.e., IBRNet and NeuRay in this work) is that their original codebases work on perspective instead of panoramic imagery (L36-38). However, adapting perspective projection to equirectangular projection is trivial. Some common adaptation like circular padding is easy to add as well. Many deep-learning based models also show promising quality by adapting CNN on equirectangular panorama like LayoutNet[Zou et al] and DuLa-Net[Yang et al], just name a few. With these simple adaptation, the baseline methods can also be directly trained on equirectangular panorama without the claimed disadvantage.

**Some design choices are not well justified and discussed.** Please refer to question 1,2,3 for detail.

**Paper writing**
- Sec.3.1 L122-123. $w_i$ is not the accumulated transmittance. Only the $\prod_{k=1}^{i-1} (1 - \alpha_k)$ is.
- The $w$ is duplicated in Eq.1 and Eq.4 but they are totally different things. I suggest using different notations.
- Table.1 and Table.2. It's better to directly use a table footnote describing the "*" means the models are trained with perspective cubemap projection and evaluated with equirectangular projection.
- L140. "When" should be "when".


**Questions:**

1. Why constraint the proposed method on aggregating only 2 panorama views (Eq.3)? The original NeuRay have tried many different number of working views and find 8 nearyby views achieves saturated quality. Experiment to justify the design choice in this work is missing.
2. The search space of the stereo depth candidacy is scene dependent and crucial. How is the $\beta$ in L215 determined? If applying the proposed method to other datasets, is there any rule of thumb to determine the $\sigma$ based on the scene scales?
3. In supp's Table.3. It's strange that the more fine-grained is the sampling (i.e., more $N_{mono}$) the worse results the proposed method can achieve. The paper should discuss about this experiment. Besides, what if we using even less $N_{mono}$? Given that the best result is achieved by the minimum $N_{mono}=5$ setting in Table.3, I think the ablation to find the optimum $N_{mono}$ is incomplete.


**Limitations:**

The discussed limitation looks reasonable.

---

> ### Author Rebuttal · Authors · 2023-08-09
>
> Dear Reviewer 74Ux:
>
> Thanks for the review and comments.  We hope the following responses could solve your concerns.
>
> * For Weakness 1:
> >Adapting from perspective cubemap to equirectangular projection...
>
>      We illustrate the key differences between PanoGRF and NeuRay from many aspects in the "global" response. Please refer to the more detailed clarification in our global response.
>
>     The novel view synthesis task for wide-baseline panoramic images is challenging. First, input views are sparse. Second, the occlusion issue is severe. To tackle the first challenge, we develop generalizable spherical radiance fields that incorporate 360&deg; data priors into the spherical NeRF. For the second challenge, we introduce 360&deg; monocular depth into PanoGRF.
>
>     We build up PanoGRF based on our key observations. NeuRay can only work on perspective images. If the perspective slices of the panoramic image are input, incorrect features are easily aggregated during the feature aggregation process due to the limited field of view of the perspective image. To solve this problem, we use spherical projection for feature alignment, maintaining a spherical image representation. Secondly, NeuRay relies on planar depth, while PanoGRF relies on spherical depth. Spherical depth can provide more robust geometric features, avoid feature alignment issue similar to that mentioned earlier, and offer a full field of view and better continuity. We also observed that ordinary 360&deg; MVSNet based on merely multi-view matching cannot handle the serious occlusion problem under a wide baseline. We use 360 monocular depth to guide the deep sampling process of spherical sweeps to introduce strong single-view priors. Furthermore, our method is simple and highly effective.
>
>     Adapting perspective to equirectangular is not trival. Many papers, such as [1, 2], worked on this. Besides, we also designed the additional targeted methods for improvements, as mentioned above. We also have attempted to improve the performance of CNN in PanoGRF to alleviate spherical distortion by introducing part of the methods [1, 2]. But these methods make almost no difference and increase the complexity of PanoGRF for 360&deg; view synthesis task. What truly works in the synthesis of wide-baseline panoramic views are our proposed solutions.
>
> * For Weakness 2:
> >...Please refer to question 1,2,3 for detail.
>
>     We answer your questions in subsequent paragraphs.
>
> * For Weakness 3:
> >Paper writing...
>
>     Thanks for your correction. According to your suggestions, we will correct these mistakes in the main body of our paper.
>
> * For Question 1:
> >Why constraint the proposed method on aggregating only 2 panorama views...
>
>     We utilize two wide-baseline panoramic views as inputs because our research focuses on the camera baseline size rather than the number of panoramic images. Therefore, we conducted comparative experiments under varying camera baselines.
>
>     However, our method is not limited to two panoramas. To further verify the effectiveness of our method, we conducted comparative experiments with NeuRay using multiple panoramic image as inputs. We placed the four input viewpoints at the corners of a horizontal square and tested the rendering performance at the center viewpoint, as well as other viewpoints located at -0.4, -0.2, -0.1, -0.05, 0.05, 0.1, 0.2, and 0.4 meters in the vertical direction from the center viewpoint. The diagonal length of the square is 1.0 meters. The table below presents the quantitative comparison results. As shown, PanoGRF still largely outperforms NeuRay with multiple panoramic inputs. The qualitative results can be found in the attached PDF of the global response.
>
>     |method|PSNR|WS-PSNR|SSIM|LPIPS|
>     |:----:|:----:|:----:|:----:|:----:|
>     |NeuRay|27.82|26.74|0.8614|0.2312|
>     |PanoGRF|**28.99**|**27.91**|**0.8762**|**0.2071**|
>
> * For Question 2:
> >...How is the \beta in L215 determined...
>
>
>     $\beta$ is taken to be 3 because [$\mu$-3*$\sigma$, $\mu$+3*$\sigma$] is a frequently used interval in Gaussian sampling in the vicinity of the expectation value.
>
>     The value $\sigma$=0.5 is obtained from experiments on Matterport3D, where we tested the values 0.1, 0.5, 1.0, and 1.5, with 0.5 proving to be the most effective. In general cases, this value does not require adjustment for indoor scenes. However, if the depth scale of a scene is particularly large, this value may need adaptation. For instance, $\sigma$ can be set as 1/20 of the maximum depth of the scene.
>
> * For Question 3:
> >In supp's Table.3. It's strange that ...
>
>     Thanks for your suggestion! We added the experiment as follows:
>
>     |$N_{mono}$|L1|L2|RMSE|WS-L1|WS-L2|WS-RMSE|
>     |:----:|:----:|:----:|:----:|:----:|:----:|:----:|
>     |1|0.1586|0.2498|0.4317|0.1745|0.1971|0.3937|
>     |3|**0.1432**|**0.1993**|**0.3865**|0.1529|0.1649|0.3580|
>     |5|0.1441|0.2047|0.3877|**0.1502**|**0.1624**|**0.3546**|
>     |7|0.1496|0.2252|0.4104|0.1640|0.1896|0.3832|
>     |9|0.1645|0.2912|0.4511|0.1752|0.2215|0.4059|
>
>     The reliability of 360&deg; monocular depth estimation is not perfect. Therefore, our paper employs uniform sampling to compensate for the remaining depth candidates. In the ablation experiments, we consistently maintain $N_{mono}+N_{uni}=64$, where $N_{uni}$ represents the sample number of uniform distribution. We discovered that the configuration $N_{mono}=5$ yields the best results with regard to the metrics of WS-L1, WS-L2, and WS-RMSE.
>
> Sincerely,
>
> Authors
>
> [1] Jiang, H., Sheng, Z., Zhu, S., Dong, Z., & Huang, R. (2021). Unifuse: Unidirectional fusion for 360 panorama depth estimation. IEEE Robotics and Automation Letters, 6(2), 1519-1526.
>
> [2] Li, Y., Guo, Y., Yan, Z., Huang, X., Duan, Y., & Ren, L. (2022). Omnifusion: 360 monocular depth estimation via geometry-aware fusion. In Proceedings of the IEEE/CVF Conference on Computer Vision and Pattern Recognition (pp. 2801-2810).

---

> > ### Author Response · Authors · 2023-08-18
> > **Would you give us a response?**
> >
> > Dear Reviewer 74Ux:
> >
> > We are sorry to bother you. We sincerely thank you for the review and comments. We have provided corresponding responses and results, which we believe have covered your concerns.
> >
> > As the author-reviewer discussion is approaching the deadline, we hope to receive your response before the deadline. If you have any other questions, we are willing to discuss them with you at any time.
> >
> > Sincerely,
> >
> > Authors

---

> > > ### Comment · Reviewer_74Ux · 2023-08-19
> > >
> > > Sorry for the delayed reply. I sincerely appreciate the author's effort in the response, which addressed my questions about the technical aspects.
> > >
> > > However, my main concerns about the novelty and limited contribution still remain.
> > > In the global response section:
> > > - **Feature aggregation.** The mentioned main limitation of existing work is that they use perspective projection and the proposed work adapt to spherical projection. To me, it is a pure engineering effort without meaningful new insight.
> > > - **Depth estimation.** The main proposal is to use 360 depth estimation models instead of the previously used perspective monocular depth prior. Similar concerns as mentioned above apply. Simply changing a pre-trained model may not suffice.
> > >
> > > Overall, I'm not convinced that the proposed work presents enough technical novelty or new insight beyond an engineering effort. I believe the authors could attempt to change my opinions by addressing at least one of the following two aspects:
> > > 1. I agree adapting perspective methods to 360 by **incorporating novel domain-specific insight** is worth doing. Like the employed UniFuse design their model to take advantage of different projections of 360 images. The support for spherical projection and the utilization of different pre-trained models appear to be primarily engineering efforts without introducing new insights. This, in my view, may not meet the standards expected at NeurIPS.
> > > 2. This work attempts to address two primary challenges: (1) handling sparse input views and (2) using a monocular depth prior to tackle occlusion. However, there are already numerous existing methods for dealing with sparse views (e.g., [a,10,13,18,b]) and leveraging depth prior (e.g., [c,d,e]). **What new merits does the proposed method offer compared to related methods when addressing these challenges?** Does the proposed method yield superior results? Does it leverage any general or domain-specific priors to achieve better outcomes? Merely altering the input image projection or the choice of the pre-trained network may not be sufficient.
> > >
> > > [a] Chen et al., MVSNeRF: Fast Generalizable Radiance Field Reconstruction from Multi-View Stereo, ICCV 2021.
> > > [b] Chen et al., GeoAug: Data Augmentation for Few-Shot NeRF with Geometry Constrains, ECCV 2022.
> > > [c] Roessle et al., Dense Depth Priors for Neural Radiance Fields from Sparse Input Views, CVPR 2022.
> > > [d] Deng et al., Depth-supervised NeRF: Fewer Views and Faster Training for Free, CVPR 2022.
> > > [e] Yu et al., MonoSDF: Exploring Monocular Geometric Cues for Neural Implicit Surface Reconstruction, NeurIPS 2022.

---

> > > > ### Comment · Reviewer_zV2U · 2023-08-20
> > > > **Official Comment by Reviewer zV2U**
> > > >
> > > > I think the concerns of reviewer 74Ux are necessary. I hope the author can give some comparison with other related methods. A potential naive experiment may be using the related method with the sparse mono input, and then transforming the result to the panorama. If the proposed method PanoGRF outperforms the naive solution, it will demonstrate the insight of the proposed method well. Please consider it.

---

> > > > > ### Author Response · Authors · 2023-08-20
> > > > >
> > > > > Dear Reviewer zV2U:
> > > > >
> > > > > We appreciate your suggestions and positive comments. We sincerely hope our response can cover your concerns.
> > > > >
> > > > > >A potential naive experiment may be using the related method with the sparse mono input, and then transforming the result to the panorama. If the proposed method PanoGRF outperforms the naive solution, it will demonstrate the insight of the proposed method well.
> > > > >
> > > > > **We have already included the comparisons with existing methods using sparse monocular input and then rendering the result into panoramas in the main paper and the supplementary.** PanoGRF outperforms the existing methods mentioned due to its spherical representation and incorporation of 360-degree monocular depth.
> > > > >
> > > > > - MVSNeRF[a] is not as effective as NeuRay[f], as demonstrated in the comparative experiments of NeuRay. NeuRay* in Sec.4.3 of the main text of our paper takes cubemaps (i.e., sparse monocular perspective images) and corresponding planar depths as input. We evaluated the panoramic results of NeuRay*. Extensive experiments in Sec.4.3 have shown that PanoGRF is superior to NeuRay* due to the spherical representation. The inherent limitations of cubemap projections (perspective projections), such as diminished continuity and restricted field of view, affect both feature aggregation and depth estimation.
> > > > > - All the mentioned methods except MVSNeRF are per-scene optimized. Under wide-baseline settings, per-scene optimization methods with depth priors (e.g., Dense Depth Priors [c]) lead to overfitting, as demonstrated in Sec. E of the supplementary material and highlighted by Reviewer `WLFo`. It's more robust to train a generalizable renderer under the wide-baseline setting. The direct comparisons with Dense Depth Priors using cubemaps inputs, should be consistent with that of supplementary materials, and we will update the experimental results if time allows.
> > > > >
> > > > > Thanks again for your response and positive feedback.
> > > > >
> > > > > Sincerely,
> > > > >
> > > > > Authors
> > > > >
> > > > >
> > > > >
> > > > > [a] Chen et al., MVSNeRF: Fast Generalizable Radiance Field Reconstruction from Multi-View Stereo, ICCV 2021.
> > > > >
> > > > > [b] Chen et al., GeoAug: Data Augmentation for Few-Shot NeRF with Geometry Constrains, ECCV 2022.
> > > > >
> > > > > [c] Roessle et al., Dense Depth Priors for Neural Radiance Fields from Sparse Input Views, CVPR 2022.
> > > > >
> > > > > [d] Deng et al., Depth-supervised NeRF: Fewer Views and Faster Training for Free, CVPR 2022.
> > > > >
> > > > > [e] Yu et al., MonoSDF: Exploring Monocular Geometric Cues for Neural Implicit Surface Reconstruction, NeurIPS 2022.
> > > > >
> > > > > [f] Liu, Yuan, et al. "Neural rays for occlusion-aware image-based rendering." Proceedings of the IEEE/CVF Conference on Computer Vision and Pattern Recognition. 2022.

---

> > > > ### Author Response · Authors · 2023-08-20
> > > >
> > > > Dear Reviewer 74Ux:
> > > >
> > > > First and foremost, we appreciate the time and effort you've invested in reviewing our paper. While we respect your perspective and opinions on the paper, we must express our disagreement with your evaluation concerning the novelty and overall contribution of our work. In the following response, we will clarify our position, providing evidence from the paper and the rebuttal, highlighting the novel aspects and the significance of the contributions we believe our work brings to the domain.
> > > >
> > > > ---
> > > > >**Feature aggregation.** The mentioned main limitation of existing work is that they use perspective projection and the proposed work adapt to spherical projection. To me, it is a pure engineering effort without meaningful new insight.
> > > >
> > > > >**Depth estimation.** The main proposal is to use 360 depth estimation models instead of the previously used perspective monocular depth prior. Similar concerns as mentioned above apply. Simply changing a pre-trained model may not suffice.
> > > >
> > > > >The support for spherical projection and the utilization of different pre-trained models appear to be primarily engineering efforts without introducing new insights.
> > > >
> > > > We disagree with this comment. **PanoGRF does not simply adapt the spherical projection and replace a pre-trained depth estimation model.**
> > > >
> > > > 1. **PanoGRF offers three distinctive insights for spherical novel view synthesis using wide-baseline panoramas.**
> > > >
> > > > - **Using spherical/panorama representation is better than cubemaps because it offers a full field of view.** Extensive comparative experiments in Sec. 4.3 have verified that handling spherical representation is much more effective than using cubemaps (please refer to the comparisons with IBRNet* and NeuRay*). We observed that when using common perspective view synthesis methods, these approaches encounter a significant issue during feature aggregation. Due to the limited field of view of the perspective view, a 3D sample point is often projected outside the perspective view or behind the camera (z-depth < 0). Aggregating features in this manner is incorrect. PanoGRF leverages the spherical representation to capitalize on the full field-of-view inherent in panoramic images. This approach involves aggregating appearance and geometry features in a spherical manner. Cubemap projections (perspective projections) inherently exhibit drawbacks, such as reduced continuity and limited field of view, which impact both feature aggregation and depth estimation.
> > > >
> > > > - **It is essential to train a robust generalizable model for novel view synthesis using wide-baseline panoramas.** Under wide-baseline settings, per-scene optimization methods with depth priors (e.g., Dense Depth Priors [c]) lead to overfitting, as demonstrated in Sec. E (fine-tuning of PanoGRF) of the supplementary material and highlighted by Reviewer `WLFo`.
> > > >
> > > > - **The quality of the spherical depth is crucial for training the generalizable model and achieving high-quality novel view synthesis results.** Conventional 360-degree MVSNet struggles to effectively address occlusion issues. We propose an innovative solution to mitigate the occlusion challenge by utilizing 360-degree monocular depth to guide spherical depth sampling in 360-degree MVSNet, thereby enhancing the quality of spherical depth. Ablation studies in Sec. 4.4 of the main text confirms that 360° MVSNet and 360° monocular Depth, despite their individual limitations, complement each other and play a vital role in 360° novel view synthesis under wide baselines.
> > > >
> > > > 2. **PanoGRF does not simply replace a pre-trained depth estimation model.** As we mentioned in the last paragraph, naive 360-degree MVSNet cannot handle occlusion issues by merely using multi-view matching under a wide-baseline setting. We incorporate 360-degree monocular depth into 360-degree MVSNet to provide strong single-view priors. Additionally, spherical depth is integral to PanoGRF's spherical design, offering better continuity and a full field of view compared to planar depth predicted from cubemaps.
> > > >
> > > > To be continued...

---

> > > > > ### Author Response · Authors · 2023-08-20
> > > > >
> > > > > ---
> > > > > >However, there are already numerous existing methods for dealing with sparse views (e.g., [a,10,13,18,b]) and leveraging depth prior (e.g., [c,d,e]). *What new merits does the proposed method offer compared to related methods when addressing these challenges? Does the proposed method yield superior results? Does it leverage any general or domain-specific priors to achieve better outcomes?*
> > > > >
> > > > >
> > > > > 1. **New merits and domain-specific priors.**
> > > > >
> > > > > - The new merits:
> > > > >
> > > > >     a) PanoGRF uses spherical/panorama representation, which is better than cubemaps because it offers a full field of view.
> > > > >
> > > > >     b) It is also essential to train a robust generalizable model for novel view synthesis using wide-baseline panoramas.
> > > > >
> > > > > - The domain-specific priors:
> > > > >     Under wide-baseline setting, the quality of the spherical depth is crucial for training the generalizable model and achieving high-quality novel view synthesis results. 360-degree MVSNet struggles to effectively address occlusion issues. PanoGRF utilizes 360-degree monocular depth to guide spherical depth sampling in 360-degree MVSNet, thereby enhancing the quality of spherical depth.
> > > > >
> > > > > 2. **Superior results.**
> > > > >
> > > > >
> > > > >     PanoGRF outperforms the existing methods you mentioned due to its spherical representation and incorporation of 360-degree monocular depth as mentioned above.
> > > > >
> > > > >     - **MVSNeRF[a] suffers similar issues to NeuRay** and it underperforms compared to NeuRay, as evidenced by the comprehensive comparative experiments conducted in NeuRay.
> > > > >     - **Other methods you mentioned are per-scene optimized.** Per-scene optimization with depth priors under the wide-baseline setting leads to overfitting as estimated depths are not perfect due to the severe occlusion issue.  In Sec. E (fine-tuning of PanoGRF) of the supplementary materials, we demonstrate that per-scene optimization with depth supervision, such as Dense Depth Priors[c], yields inferior performance compared to general renderers. Furthermore, MonoSDF[e] relies solely on monocular cues, which are inadequate for wide-baseline settings. Our ablation studies confirm that both 360-degree monocular depth and 360-degree MVSNet are crucial for 360-degree novel view synthesis task. InfoNeRF[13], RegNeRF[18], GeoAug[b] and DSNeRF[d] do not learn any scene priors from datasets. DietNeRF[10] employs semantic consistency loss to enhance the quality of novel views, which is less effective than direct methods for improving geometry, such as [13, 18].
> > > > >
> > > > > ---
> > > > > In summary, PanoGRF has its new merits and unique insights for the wide-baseline 360-degree view synthesis task. We sincerely hope our responses can resolve your concerns. Thanks again for your time and effort.
> > > > >
> > > > >
> > > > > Best regards,
> > > > >
> > > > > Authors.
> > > > >
> > > > > [a] Chen et al., MVSNeRF: Fast Generalizable Radiance Field Reconstruction from Multi-View Stereo, ICCV 2021.
> > > > >
> > > > > [b] Chen et al., GeoAug: Data Augmentation for Few-Shot NeRF with Geometry Constrains, ECCV 2022.
> > > > >
> > > > > [c] Roessle et al., Dense Depth Priors for Neural Radiance Fields from Sparse Input Views, CVPR 2022.
> > > > >
> > > > > [d] Deng et al., Depth-supervised NeRF: Fewer Views and Faster Training for Free, CVPR 2022.
> > > > >
> > > > > [e] Yu et al., MonoSDF: Exploring Monocular Geometric Cues for Neural Implicit Surface Reconstruction, NeurIPS 2022.

---

### Official Review · Reviewer_kb7G · 2023-07-04

**Soundness:** 3 good
**Presentation:** 4 excellent
**Contribution:** 1 poor
**Rating:** 4
**Confidence:** 5

**Summary:**

This paper presents a method called PanoGRF for synthesizing novel panoramas using two wide-baseline panoramas, with the incorporation of 360 scene priors into Spherical NeRF to generate new views. The method involves extracting appearance and geometry features from the input panoramas and estimating spherical depths through convolutions. To enhance the accuracy of 360 panoramas MVS depth estimation, the authors integrated a 360 monocular depth estimation network. Specifically, they employed a Gaussian distribution to sample depth candidates around the estimated 360 monocular depth. Extensive experiments were conducted to demonstrate the effectiveness of the proposed method in synthesizing novel views from wide-baseline panorama stereos.

**Strengths:**

The authors have performed thorough experiments to showcase the effectiveness of the proposed method, making the study comprehensive. The results obtained from the performance evaluation and ablation study are noteworthy, highlighting that the proposed method achieves state-of-the-art performance. The contributions of the proposed components in achieving this success are significant.

Moreover, the paper is excellently written and presents the concepts in a clear and easily understandable manner. The figures are well-drawn, effectively conveying the main ideas and providing visual clarity to the readers.

**Weaknesses:**

The level of novelty in this paper is relatively limited since many of the components utilized have been proposed by previous methods. For instance, the appearance and geometry feature aggregation techniques are adopted from NeuRay, while the 360 MVS approach, which involves cost volume and mono-guided depth sampling, can be attributed to [14, 12, 28]. Furthermore, it is not particularly challenging to adapt these techniques to panoramas by employing spherical projection.

It is worth noting that the comparison with IBRNet and NeuRay may not be entirely fair due to the differing assumptions regarding the number of input images. Both IBRNet and NeuRay assume multiple images as inputs, while this paper solely employs two images in its experiments.

**Questions:**

Although all experiments in the paper demonstrate the method's capability to generate views along the baseline of two cameras, the design of the proposed method suggests the potential to generate views outside of these limited regions. It would be valuable to investigate how the method performs in synthesizing views beyond the baseline. By conducting additional experiments to test the boundary and assess the performance of the proposed method in generating views outside the baseline, a more comprehensive understanding of its capabilities can be gained. This would be a beneficial addition to further validate and explore the boundaries of the proposed method.

**Limitations:**

The authors have mentioned the limitations and societal impact of the paper in lines 273-278. It well covered the possible limitations and potential societal impact of the paper.

---

> ### Author Rebuttal · Authors · 2023-08-08
>
> Thanks for your effort for reviewing our paper. We hope the following responses could solve your major concerns.
>
> * For weakness 1:
> >The level of novelty...
>
>
>     PanoGRF is not a simple combination of NeuRay and the 360&deg; novel view synthesis task. For a more detailed clarification, please refer to our global response.
>
>     The novel view synthesis task for wide-baseline panoramic images is indeed highly challenging. First, input views are sparse. Second, the occlusion issue is severe. To address the first challenge, we develop generalizable spherical radiance fields that incorporate 360&deg; data priors into the spherical NeRF, preventing the overfitting problem. For the second challenge, we introduce 360&deg; monocular depth to guide the spherical depth sampling of 360&deg;MVSNet.
>
>     It is also important to note that PanoGRF is not simply the combination of NeuRay and spherical projection. We built PanoGRF based on our key observations. As we metioned in the global response, NeuRay can only work on perspective images. If the perspective slices of the panoramic image are input, incorrect features are easily aggregated during the feature aggregation process due to the limited field of view of the perspective image. To solve this problem, we use spherical projection for feature alignment, maintaining a spherical image representation, and fully leveraging the omnidirectional field of view provided by panoramas. Secondly, NeuRay relies on planar depth, while PanoGRF relies on spherical depth. For generalizable renderers of panoramic images, spherical depth can provide more robust geometric features, avoid feature alignment issue similar to that mentioned earlier, and offer a full field of view and better continuity. We also observed that ordinary 360MVSNet based on merely multi-view matching cannot handle the serious occlusion problem under a wide baseline. In this regard, we proposed a feasible solution. Specifically, we use 360 monocular depth to guide the deep sampling process of spherical sweeps to introduce strong single-view priors.
>
>     Furthermore, our method is simple and highly effective, as demonstrated in the extensive comparative experiments (Sec.4.3 of the main text, Sec.E and Sec.F of the supplementary materials). These experiments validate that our solutions are crucial for wide-baseline panoramas.
>
> * For Weakness 2:
> >...the comparison with IBRNet and NeuRay may not be entirely fair...
>
>      We would like to clarify that we utilize two wide-baseline panoramic views as inputs because our research focuses on the camera baseline size rather than the number of panoramic images. By ensuring that all experiments have two panoramic image inputs, we facilitate the execution of the experiments. Correspondingly, we split each panorama into cubemaps, with six sides per panorama, and then use all the cubemaps as inputs for NeuRay/IBRNet. Consequently, NeuRay/IBRNet receives twelve perspective images as inputs. This comparison remains fair under these input conditions.
>
>     However, our method is not limited to two panoramas. For instance, when rendering a test view in the renderer module, we use $N$ input panoramas as reference images, and the renderer does not need to be modified. In the 360&deg; spherical depth estimator module, for each reference image, we use the other $N-1$ input panoramas as source images. We average the multiple cost-volumes obtained during 360&deg; multi-view matching process between the reference image and each source image. The rest is unchanged. In this way, our method can be applied to the multi-view panoramic inputs.
>
>
>     To further verify the effectiveness of our method, we conducted comparative experiments with NeuRay using multiple panoramic image as inputs. We placed the four input viewpoints at the corners of a horizontal square and tested the rendering performance at the center viewpoint and other viewpoints located at -0.4, -0.2, -0.1, -0.05, 0.05, 0.1, 0.2, and 0.4 meters in the vertical direction from the center viewpoint. The diagonal length of the square is 1.0 meters. The table below presents the quantitative comparison results between PanoGRF and NeuRay. As shown, PanoGRF still largely outperforms NeuRay with multiple panoramic inputs. The qualitative results can be found in the attached PDF of the global response.
>
>
>     |method|PSNR|WS-PSNR|SSIM|LPIPS|
>     |:----:|:----:|:----:|:----:|:----:|
>     |NeuRay|27.82|26.74|0.8614|0.2312|
>     |PanoGRF|**28.99**|**27.91**|**0.8762**|**0.2071**|
>
>
>
> * For Question:
> >...how the method performs in synthesizing views beyond the baseline...
>
>     Thanks for your valuable suggestion! We conducted an additional experiment where novel views were generated at positions 0.25 and 0.5 meters above the middle point between two input viewpoints. The input viewpoints are 1.0 meters apart.  We compared the quantitative results of our method with NeuRay's as shown in the table below. PanoGRF consistently surpassed NeuRay's performance, indicating its capacity to yield superior results beyond the camera baseline. For the qualitative comparison, please refer to the attached PDF file of our "global" rebuttal. We also present the failure case of PanoGRF under such condition in the attached PDF file. In some viewpoints, a previously occluded area may be visible and since this area has not been seen in either of the two existing viewpoints, the synthesis of this area is not effective. The area, which has not been seen in either of the existing viewpoints may be able to be filled in by combining with the existing diffusion generative approach. This is the next direction we plan to investigate.
>
>
>     |distance|method|PSNR|WS-PSNR|SSIM|LPIPS|
>     |:----:|:----:|:----:|:----:|:----:|:----:|
>     |0.25m|NeuRay|20.66|20.05|0.714|0.409|
>     |0.25m|PanoGRF|**21.98**|**21.30**|**0.763**|**0.348**|
>     |0.5m|NeuRay|20.39|19.94|0.711|0.420|
>     |0.5m|PanoGRF|**21.99**|**21.42**|**0.769**|**0.349**|
>
> Best,
>
> Authors

---

> > ### Author Response · Authors · 2023-08-18
> > **Would you give us a response?**
> >
> > Dear Reviewer kb7G:
> >
> > We are sorry to bother you. We sincerely thank you for the review and comments. We have provided corresponding responses and results, which we believe have covered your concerns.
> >
> > As the author-reviewer discussion is approaching the deadline, we hope to receive your response before the deadline. If you have any other questions, we are willing to discuss them with you at any time.
> >
> > Sincerely,
> >
> > Authors

---

> > ### Comment · Reviewer_kb7G · 2023-08-20
> >
> > I gratefully acknowledge the author's comprehensive response, which addressed several of my concerns regarding the comparison and performance beyond the camera baseline. After a thorough examination of the feedback from other reviewers, I keep my original rating that the manuscript lacks the requisite novelty for a NeurIPS publication.
> >
> > The predominant components of the network and feature design derive from prior work. The principal contribution of this manuscript lies in adapting these features/networks from perspective projection to spherical projections, a sentiment echoed by reviewer 74Ux. While the authors undoubtedly demonstrate a commendable effort, enhancing the efficacy of preceding networks on panoramas, the degree of novelty presented remains circumscribed and falls short of the standards set for NeurIPS publication.

---

> > > ### Author Response · Authors · 2023-08-20
> > >
> > > Dear Reviewer kb7G,
> > >
> > > Thank you for your additional feedback. We respect your viewpoint, yet we believe there might be some misunderstandings or underestimations regarding the novelty and significance of our work. To address your concerns:
> > >
> > > > The predominant components of the network and feature design derive from prior work. The principal contribution of this manuscript lies in adapting these features/networks from perspective projection to spherical projections.
> > >
> > > 1. **PanoGRF's Novelty and Significance**
> > >
> > >   - **Full Field of View with Spherical Representation:** We'd like to emphasize the advantage of using a spherical representation over cubemaps. As elaborated in Sec. 4.3, our comparison experiments affirm that the spherical representation surpasses cubemaps in effectiveness (please refer to the comparison with IBRNet* and NeuRay*). Common perspective view synthesis methods, when used, often face challenges during feature aggregation. This is due to the limited field of view causing projection errors. PanoGRF strategically uses the spherical representation to harness the entire field-of-view of panoramic images. This unique methodology offers enhanced feature aggregation and depth estimation, bypassing the limitations of cubemaps.
> > >
> > >   - **Importance of Robust Generalizable Models:** As also observed by Reviewer WLFo, there is a pressing need for models that can withstand overfitting in wide-baseline settings. Our supplementary material in Section E demonstrates that techniques such as Dense Depth Priors [c], when employed in per-scene optimization, tend to overfit. Addressing this challenge, our research emphasizes the development of a robust, generalizable model suitable for these settings.
> > >
> > >   - **Innovative Solution for Occlusion Issues:** Addressing occlusion in 360-degree MVSNet has been a significant challenge in the field. Our proposal of using 360-degree monocular depth to guide the spherical depth sampling introduces an innovative solution to this problem. Ablation studies, as referenced in Sec. 4.4, reveal that combining 360° MVSNet with 360° monocular Depth optimally leverages their strengths, producing superior synthesis results.
> > >
> > > 2. **Beyond Simple Adaptation**
> > >
> > >   - Our methodology is not a mere adaptation from perspective projection to spherical projections. It's essential to note that a naive 360-degree MVSNet remains insufficient in handling occlusion issues, especially in wide-baseline scenarios. PanoGRF's design is integrative—it brings together 360-degree monocular depth and 360-degree MVSNet, bolstered with single-view priors. Plus, by employing spherical representation, PanoGRF maintains the integrity of panoramic images, offering an improved continuity and comprehensive view than traditional methods.
> > >
> > > To conclude, we firmly believe that PanoGRF offers novel methodologies and insights into the realm of wide-baseline spherical view synthesis. We trust this response sheds light on our perspective and hope it addresses your concerns adequately. We are grateful for your thorough evaluation and welcome any further feedback.
> > >
> > > Sincerely,
> > >
> > > Authors

---

### Official Review · Reviewer_un5g · 2023-07-05

**Soundness:** 3 good
**Presentation:** 3 good
**Contribution:** 2 fair
**Rating:** 6
**Confidence:** 4

**Summary:**

This paper introduces PanoGRF, a method for generalizable novel view synthesis of sparse panorama images with wide baselines. This PanoGRF is basically built upon the perspective view synthesis method, NeuRay. Previous generalizable NeRF methods are mainly designed for perspective images are may introduce extra errors between the projection of perspective views and panorama views.

To perform generalizable view synthesis given multiple panoramas, the authors 1) design CNN modules on panoramas to extract the geometry and appearance fields, constructing the cost volumes for density and color estimation; 2) design a 360-panorama depth estimation method based on both multi-view image feature correlations and cues of monocularly estimated depth maps. The predicted depth maps are for the upcoming geometry feature extraction.



**Strengths:**

- The paper is well written and easy to understand.
- Lots of supplementary materials are given to help the readers fully appreciate the experimental results and algorithm designs.
- The proposed method demonstrates state-of-the-art view synthesis quality on generalizable view synthesis task from panorama images.

**Weaknesses:**

- The overall idea for generalizable view synthesis is not new and is basically built upon the existing NeuRay work, therefore it is a bit incremental for the research community.
- The qualitative results are limited in indoor scenes as discussed in the limitations. It would be better if PanoGRF can be trained and evaluated on outdoor unbounded scenes.

**Questions:**

The paper is written very clearly, and I find no significant questions to ask.


**Limitations:**

The limitations are adequately discussed.

---

> ### Author Rebuttal · Authors · 2023-08-08
>
> Thank you for the review and comments. We are pleased to receive positive feedback on our performance and writing. We hope the following responses will address your concerns effectively.
>
> * For Weakness 1:
> >The overall idea for generalizable view synthesis is not new and is basically built upon the existing NeuRay work, therefore it is a bit incremental for the research community.
>
>     We think it is actually not incremental. Please refer to the detailed contribution clarification in our global response.
>
>     Novel view synthesis from wide baseline panoramas is a very challenging task. As we mentioned in the "global" rebuttal, both NeRF and perspective generalization methods cannot achieve good results under such settings. It is non-trivial to directly apply perspective methods to wide-baseline panoramas because of different 3D representation and severe occlusion problem. We observed their limitations and proposed crucial solutions: we no longer rely on ordinary perspective inputs and planar depth (z-depth), but directly input panoramic images and spherical depth. We alignment appearance feature and geometry feature in the spherical representation. To provide a more robust geometry feature, we introduce 360&deg; monocular depth into 360&deg; MVSNet further increasing the upper limit of PanoGRF. PanoGRF is inspired by these perspective generalization methods but it is not incremental.  The comprehensive experiments validate that our solutions are vital for wide-baseline panoramas. More importantly, no similar work has been done on panoramic images.
>
> Novel view synthesis from wide-baseline panoramas is an extremely challenging task. As mentioned in our global" response, both NeRF and perspective generalization methods struggle to achieve good results under such settings. Directly applying perspective methods to wide-baseline panoramas is non-trivial due to differences in 3D representation and the severe occlusion problem. We observed their limitations and proposed crucial solutions: instead of relying on ordinary perspective inputs and planar depth (z-depth), we directly input panoramic images and spherical depth. We align appearance and geometry features in the spherical representation. To provide a more robust geometry feature, we introduce 360&deg; monocular depth into 360&deg; MVSNet, further increasing the upper limit of PanoGRF. Although PanoGRF is inspired by perspective generalization methods, it is not incremental. Our comprehensive experiments validate that our solutions are vital for wide-baseline panoramas. More importantly, no similar work has been done on panoramic images.
> * For Weakness2:
> >The qualitative results are limited in indoor scenes as discussed in the limitations. It would be better if PanoGRF can be trained and evaluated on outdoor unbounded scenes.
>
>     Indeed, the current absence of large-scale outdoor panoramic datasets prevents PanoGRF from being trained on outdoor scenes. Furthermore, the significant depth scale differences between indoor and outdoor settings limit PanoGRF's generalization capabilities for outdoor scenes, which also applies to other generalizable methods for perspective images. But if large-scale outdoor panoramic dataset is available, PanoGRF can also be trained and evaluated on outdoor scenes. We consider addressing this issue as part of our future work.
>
> Sincerely,
>
> Authors

---

> > ### Author Response · Authors · 2023-08-18
> > **Would you give us a response?**
> >
> > Dear Reviewer un5g:
> >
> > We are sorry to bother you. We sincerely thank you for the review and comments. We have provided corresponding responses and results, which we believe have covered your concerns.
> >
> > We noticed that half of the designated author-reviewer discussion period has elapsed. We hope to receive your response before the deadline. If you have any other questions, we are willing to discuss them with you at any time.
> >
> > Sincerely,
> >
> > Authors

---

> > > ### Comment · Reviewer_un5g · 2023-08-21
> > > **Comment by Reviewer un5g**
> > >
> > > Thanks for the authors' responses. The rebuttal has effectively addressed my concerns. I believe the overall quality of the paper is sufficient for acceptance.  After carefully reading the complete discussion from the other reviewers, I am aware that there is a debate regarding the technical novelty. Personally, I believe this paper attempts to achieve a novel form of generalized view synthesis on panoramas with wide baselines. This task, in my opinion, is innovative enough. Therefore, I will maintain my original rating.

---

> > > > ### Author Response · Authors · 2023-08-21
> > > >
> > > > Dear Reviewer un5g:
> > > >
> > > > We sincerely appreciate the time and effort you have dedicated to reviewing our rebuttal and all the comments, and we are grateful for your acknowledgment of our work's innovative aspects. We also believe that PanoGRF can provide inspiration and value to the community, particularly in panorama-related research, such as 3D reconstruction using panoramic inputs or multi-viewpoint panorama generation.
> > > >
> > > > Best regards,
> > > >
> > > > Authors

---

### Official Review · Reviewer_zV2U · 2023-07-05

**Soundness:** 3 good
**Presentation:** 3 good
**Contribution:** 3 good
**Rating:** 7
**Confidence:** 3

**Summary:**

This paper presents PanoGRF, Generalizable Spherical Radiance Fields for Wide-baseline Panoramas, which introduces mono-guided 360 ◦ depth estimation and leverages each panoramic view based on spherical projection. The experiments show that the proposed method significantly outperforms state-of-the-art generalizable view synthesis methods for wide-baseline panoramas (e.g., OmniSyn) and perspective images (e.g., IBRNet, NeuRay).

**Strengths:**

## Pros
1. Novelty: It's interesting to directly aggregate geometry and appearance features of 3D sample points from each panoramic view based on spherical projection. It's novel to leverage mono-guided 360 ◦ depth estimation to improve the geometry features. Although it can be argued that it's just the combination of Neuray + mono-mvs + spherical radiance fields.
2. performance: The results are remarkable compared with the sota methods.
3. writing: it's well-written and easy to follow.

**Weaknesses:**

Cons: I do not find some major concerns. The insight is easy to follow and the experiments are adequate. But I have some subjections as follows:
1. it seems to use false citation: S-NeRF[17].
2. It may lack  some ablation studies: w/o aggregation geometry/appearance feature


**Questions:**

please see the weakness.

**Limitations:**

yes.

---

> ### Author Rebuttal · Authors · 2023-08-08
>
> Dear Reviewer zV2U:
> We sincerely appreciate your review and comments. We are delighted to receive your positive feedback and score. We hope the following responses will address your concerns effectively.
>
> 1. Weakness 1:
> > it seems to use false citation: S-NeRF[17].
>
>     We apologize for causing your misunderstanding. In fact, S-NeRF represents spherical variant of NeRF which casts rays by spherical projection to adapt to panoramic inputs as described in Sec.3.1 of our paper. So we cite the original paper of NeRF. To avoid confusion, we will change the description from "S-NeRF [17]" to "S-NeRF (spherical variant of NeRF [17])" in the revision of our paper. Thanks for pointing out this issue.
>
> 2. Weakness 2:
> > It may lack some ablation studies: w/o aggregation geometry/appearance feature.
>
>    Thank you for your suggestion. We conducted ablation studies on Matterport3D, and the results are shown in the table below. In the "w/o appearance feature" ablation study, we replaced the appearance feature vector with a zero vector to disable the appearance feature while keeping other modules unchanged. We found that the model without appearance features loses its ability to infer the color of novel views entirely, as the generalizable renderer heavily relies on appearance cues from input views. In the "w/o geometry feature" ablation study, we replaced the geometry feature vector with a zero vector to disable the geometry feature while keeping other modules unchanged. We observed that although the model can still infer normal results, its performance is significantly worse than the original (full) model. Due to the length constraints of the main text, we will include these ablation studies in the supplementary material.
>
>     ||PSNR|WS-PSNR|SSIM|LPIPS|
>     |:----:|:----:|:----:|:----:|:----:|
>     |w/o appearance|5.44|5.24|0.001|0.707|
>     |w/o geometry|26.25|25.25|0.839|0.263|
>     |full|28.10|27.12|0.876|0.195|
>
> Best,
>
> Authors

---

> > ### Comment · Reviewer_zV2U · 2023-08-20
> > **Official Comment by Reviewer zV2U**
> >
> > After reading the rebuttal and all reviews, although I'm positive about this paper, I still recommend the reviewer consider the comment from the Reviewer zV2U, meanwhile, I give a potential solution comment about how to solve the concern, please see them and try to solve it. If the experiment results perform well, I would recommend accepting this paper.

---

> > > ### Author Response · Authors · 2023-08-20
> > >
> > > Dear Reviewer zV2U:
> > >
> > > We appreciate your suggestions and positive comments. We sincerely hope our response can cover your concerns.
> > >
> > > >A potential naive experiment may be using the related method with the sparse mono input, and then transforming the result to the panorama. If the proposed method PanoGRF outperforms the naive solution, it will demonstrate the insight of the proposed method well.
> > >
> > > **We have already included the comparisons with existing methods using sparse monocular input and then rendering the result into panoramas in the main paper and the supplementary.** PanoGRF outperforms the existing methods mentioned due to its spherical representation and incorporation of 360-degree monocular depth.
> > >
> > > - MVSNeRF[a] is not as effective as NeuRay[f], as demonstrated in the comparative experiments of NeuRay. NeuRay* in Sec.4.3 of the main text of our paper takes cubemaps (i.e., sparse monocular perspective images) and corresponding planar depths as input. We evaluated the panoramic results of NeuRay*. Extensive experiments in Sec.4.3 have shown that PanoGRF is superior to NeuRay* due to the spherical representation. The inherent limitations of cubemap projections (perspective projections), such as diminished continuity and restricted field of view, affect both feature aggregation and depth estimation.
> > > - All the mentioned methods except MVSNeRF are per-scene optimized. Under wide-baseline settings, per-scene optimization methods with depth priors (e.g., Dense Depth Priors [c]) lead to overfitting, as demonstrated in Sec. E of the supplementary material and highlighted by Reviewer `WLFo`. It's more robust to train a generalizable renderer under the wide-baseline setting. The direct comparisons with Dense Depth Priors using cubemaps inputs, should be consistent with that of supplementary materials, and we will update the experimental results if time allows.
> > >
> > > Thanks again for your response and positive feedback.
> > >
> > > Sincerely,
> > >
> > > Authors
> > >
> > >
> > >
> > > [a] Chen et al., MVSNeRF: Fast Generalizable Radiance Field Reconstruction from Multi-View Stereo, ICCV 2021.
> > >
> > > [b] Chen et al., GeoAug: Data Augmentation for Few-Shot NeRF with Geometry Constrains, ECCV 2022.
> > >
> > > [c] Roessle et al., Dense Depth Priors for Neural Radiance Fields from Sparse Input Views, CVPR 2022.
> > >
> > > [d] Deng et al., Depth-supervised NeRF: Fewer Views and Faster Training for Free, CVPR 2022.
> > >
> > > [e] Yu et al., MonoSDF: Exploring Monocular Geometric Cues for Neural Implicit Surface Reconstruction, NeurIPS 2022.
> > >
> > > [f] Liu, Yuan, et al. "Neural rays for occlusion-aware image-based rendering." Proceedings of the IEEE/CVF Conference on Computer Vision and Pattern Recognition. 2022.

---

> > > > ### Comment · Reviewer_zV2U · 2023-08-22
> > > > **Official Comment by Reviewer zV2U**
> > > >
> > > > Thanks to feedback from the authors. It seems the authors have addressed the concerns well. So I keep my initial rating. I'm also glad to discuss if other reviewers have different opinions. Besides, I urge the authors to release the code if the paper is accepted.

---

> > > > > ### Author Response · Authors · 2023-08-22
> > > > > **Appreciation for response and maintaining rating**
> > > > >
> > > > > Dear Reviewer zV2U:
> > > > >
> > > > > Thank you for your response, and we are pleased that you continue to uphold your rating. Rest assured, we have organized the code and will make it publicly available at the earliest opportunity. We believe that this work will significantly contribute to panorama-related research.
> > > > >
> > > > > Warm regards,
> > > > >
> > > > > Authors

---

### Author Rebuttal · Authors · 2023-08-08

Dear Reviewers:

Thank you for acknowledging the strong performance of this work. As Reviewer un5g said, quantitatively PanoGRF shows a large improvement in rendering quality and this is also evident from the example images in the figures and the videos in the supplementary. We clarify the contributions of our paper as follows to highlight our unique insights and demonstrate that our method is not just a simple composition of exsiting techniques.

The novel view synthesis task for wide-baseline panoramic images is highly challenging. Firstly, the sparsity of input views makes it difficult for NeRF to learn the correct geometry. Secondly, certain areas of the scene may be visible from only one view and occluded in others, making it difficult to provide accurate geometry using 360&deg; multi-view stereo alone. To tackle the first challenge, we develop generalizable spherical radiance fields that incorporate 360&deg; data priors into the spherical NeRF, preventing the overfitting problem. For the second challenge, we introduce 360&deg; monocular depth to guide the spherical depth sampling of 360&deg; MVSNet.



* Perspective generalization methods (such as NeuRay) inherently has some drawbacks as they only can be applied to perspective slicings obtained from panoramas. PanoGRF is dedicated for panoramic inputs based on our key observations.

    * __feature aggregation__: We observed that when employing common perspective view synthesis methods, these approaches encounter a significant issue during feature aggregation: due to the limited field of view of the perspective view, a 3D sample point is often projected outside the perspective view or behind the camera (z-depth<0). Aggregating features in this manner is incorrect. Recognizing the limitations of perspective generalization methods led us to the idea of using spherical projection for feature alignment, maintaining a spherical image representation, and fully leveraging the omnidirectional field of view provided by panoramas. We have demonstrated the importance of this approach through comparative experiments in Sec.4.3 of the main text, Sec.E and Sec.F of the supplementary materials.

    * __depth estimation__: NeuRay relies on planar depth (z-depth) while PanoGRF relies on spherical depth. It is more robust to use spherical depth predicted by 360&deg; MVSNet than planar depth predicted by ordinary MVSNet. The process of the multi-view feature matching process in perspective images suffers from a similar feature alignment problem as mentioned earlier. Moreover, splitting panoramas into cubemaps to compute z-depth disrupts the neighborhood relationship between cubemaps, which can affect accuracy. Besides, we observed that the occlusion issue is quite severe under the wide-baseline setting. Existing 360&deg; methods [1, 2, 3, 4, 5] cannot handle wide-baseline 360&deg; depth estimation in general cases. Methods [1, 2, 3] overlooked the occlusion problem under wide baselines. Methods [4, 5] considered occlusion but utilized a special camera rig to capture wide field-of-view images, making them unsuitable for wide-baseline 360° depth estimation tasks with general 360° cameras. We propose a feasible and effective solution to address this issue, which we introduce later.


	We believe that our observations can contribute to and benefit related research within the community.
* Relying solely on 360&deg; multi-view stereo is insufficient to handle the occlusion issue. To address this problem, we employ a 360&deg; monocular depth network to guide the depth sampling of 360&deg; MVSNet during sphere sweeps and contribute to the visibility prediction of the generalizable renderer.  We conducted ablation studies with various camera baselines to verify that 360&deg; MVSNet and 360&deg; Monocular Depth, despite their individual shortcomings, can complement each other and play a crucial role in the task of 360&deg; novel view synthesis under wide baselines.




In summary, we propose easy-to-implement and highly effective solutions to the challenges we encountered, and have conducted comprehensive comparisons and ablation studies to demonstrate the effectiveness of our method. Our method significantly outperforms baseline methods (including NeuRay) on several panorama datasets. Considering that Reviewer zV2U, un5g and WLFo have all agreed that our paper has great merits, we believe our research findings are worth sharing with the community.



Best,

Authors

[1] Wang, N. H., Solarte, B., Tsai, Y. H., Chiu, W. C., & Sun, M. (2020, May). 360sd-net: 360 stereo depth estimation with learnable cost volume. In 2020 IEEE International Conference on Robotics and Automation (ICRA) (pp. 582-588). IEEE.

[2] Xie, S., Wang, D., & Liu, Y. H. (2023). OmniVidar: Omnidirectional Depth Estimation From Multi-Fisheye Images. In Proceedings of the IEEE/CVF Conference on Computer Vision and Pattern Recognition (pp. 21529-21538).

[3] Chiu, C. Y., Wu, Y. T., Shen, I., & Chuang, Y. Y. (2023). 360MVSNet: Deep Multi-View Stereo Network With 360deg Images for Indoor Scene Reconstruction. In Proceedings of the IEEE/CVF Winter Conference on Applications of Computer Vision (pp. 3057-3066).

[4] Won, C., Ryu, J., & Lim, J. (2019). Omnimvs: End-to-end learning for omnidirectional stereo matching. In Proceedings of the IEEE/CVF International Conference on Computer Vision (pp. 8987-8996).

[5] Won, C., Ryu, J., & Lim, J. (2020). End-to-end learning for omnidirectional stereo matching with uncertainty prior. IEEE transactions on pattern analysis and machine intelligence, 43(11), 3850-3862.

---

### Author Response · Authors · 2023-08-21
**Appreciation for the Review and Discussion**

Dear Reviewers, AC, SAC, and PC,

As we approach the conclusion of the discussion period, we wish to convey our profound gratitude for the review of our rebuttal and the extensive comments provided. The insights, suggestions, and feedback from all reviewers have been invaluable, and we deeply appreciate the dedication and expertise that have gone into this evaluation process. Your recognition and constructive suggestions have significantly enriched our work.

Throughout this process, our primary goal has been to sincerely and systematically address each concern raised. In particular:
- **Experiments**: We incorporated additional experiments to underscore the effectiveness of PanoGRF. This includes:
  - Ablation studies w/o geometry feature and w/o appearance feature.
  - Comparisons against the state-of-the-art method, NeuRay, both with multiple panoramic inputs and beyond the camera baseline.
  - More detailed ablation studies with varying monocular depth candidates including cases where $N_{mono}<5$.
- **Novelty**: Our approach does not simply substitute projection methods and pre-trained models, and possesses distinct innovation in two aspects:
  - **Entire Field of View with Spherical Representation**: We design generalizable spherical radiance fields that go beyond the limits of conventional perspective generalizable methods, taking full advantage of the fact that panoramas have the entire field of view.
  - **Novel Solution for Occlusion Issue**: We leverage 360-degree monocular depth to guide the spherical depth sampling of 360-degree MVSNet to harness the occlusion issue.
- **Others**: We provided detailed explanations demonstrating that the methods recently suggested by Reviewer 74Ux for comparison are indeed less effective, and PanoGRF yields superior results.

While we understand and respect the reservations some reviewers have expressed regarding the technical novelty of our paper, we remain confident in the contributions our research brings not only to the domain of 360-degree view synthesis but also to broader panorama-related challenges, including 3D reconstruction with panoramic inputs and generative models tailored for multi-viewpoint panoramas.

We genuinely welcome any further feedback or inquiries you may have as we strive to refine our research.

Warm regards,

Authors

---

### Decision · Program_Chairs · 2023-09-21

**Decision:**

Accept (poster)

**Comment:**

This paper presents PanoGRF for synthesizing novel panoramas using two wide-baseline panoramas, with the incorporation of 360-depth priors into Spherical NeRF to generate new views. The paper received mixed ratings. Reviewers kb7G and 74Ux recommend rejection due to limited novelty. In their point of view, the proposed method is simply an adaptation of NeuRay to the spherical case and uses mono-depth to mitigate the wide baseline challenge. However, reviewer zV2U thinks the novelty is enough, and reviewer un5g thinks the task of novel views synthesis from wide-baseline images itself is novel. Moreover, reviewers WLFo, un5g, and zV2U agree that the results are impressive. The AC read the paper and discussion carefully. The proposed method is indeed tackling a very practical task with little prior work. Hence, it is beneficial to both the research and industry community. The qualitative and quantitative results of the proposed method are also impressive. Hence, AC recommends acceptance.